


# Anomalous winter snow amplified earthquake induced disaster of the 2015 Langtang avalanche in Nepal

Koji Fujita[1], Hiroshi Inoue[2], Takeki Izumi[3], Satoru Yamaguchi[4], Ayako Sadakane[5], Sojiro Sunako[1], Kouichi Nishimura[1], Walter W. Immerzeel[6,7], Joseph M. Shea[7], Rijan B. Kayashta[8], Takanobu Sawagaki[9], David F. Breashears[10], Hiroshi Yagi[11], Akiko Sakai[1]

[1]Graduate School of Environmental Studies, Nagoya University, Nagoya, Japan
[2]Disaster Risk Research Unit, National Research Institute for Earth Science and Disaster Prevention, Tsukuba, Japan
[3]Graduate School of Urban Environmental Sciences, Tokyo Metropolitan University, Hachioji, Japan
[4]Snow and Ice Research Center, National Research Institute for Earth Science and Disaster Resilience, Nagaoka, Japan
[5]Langtang Plan, Kamakura, Japan
[6]Department of Physical Geography, Faculty of Geosciences, Utrecht University, Utrecht, Netherlands
[7]International Centre for Integrated Mountain Development, Khumaltar, Kathmandu, Nepal
[8]School of Science, Kathmandu University, Kathmandu, Nepal
[9]Faculty of Social Sciences, Hosei University, Tokyo, Japan
[10]GlacierWorks, Marblehead, MA, USA
[11]Faculty of Education, Art and Science, Yamagata University, Yamagata, Japan

*Correspondence to*: Koji Fujita (cozy@nagoya-u.jp)

**Abstract.** Co-seismic avalanches and rock falls, and their simultaneous air blasts, which were induced by the 2015 Gorkha earthquake in Nepal, destroyed the village of Langtang. In order to reveal volume and structure of the deposit covering the village, and sequence of the multiple events, we conducted an intensive in-situ observation in October 2015. Multi-temporal digital elevation models created from photographs taken by helicopter and unmanned aerial vehicles reveal that the deposit volumes of the primary and succeeding events were $6.81 \times 10^6$ m$^3$ and $0.84 \times 10^6$ m$^3$, respectively. Visual investigations of the deposit and witness statements of villagers suggest that the primary event was an avalanche composed mostly of snow, while the contribution of collapsed glacier ice could account for a few percent of the total mass. Succeeding events were multiple rock falls which may have been triggered by aftershocks. From the initial deposited volume and the upper catchment area, we estimate an average snow depth of 1.56 m in the source area using density assumptions of snow and ice. This is consistent with anomalously large snow depths (1.28-1.52 m) observed at a neighboring glacier (4800-5100 m a.s.l.), which accumulated over the course of four major snowfall events since October 2014. Considering long-term observational data, probability density functions, and elevation gradients of precipitation, we conclude that this anomalous winter snow was an extreme event with a return interval of at least 100 years, which amplified or even caused the disaster induced by the 2015 Gorkha earthquake in Nepal.

## 1 Introduction

The avalanches and rock falls in Langtang Valley, about 70 km north from Kathmandu, resulted in one of the most remarkable tragedies in Nepal triggered by the 25 April 2015 Gorkha earthquake. Co-seismic avalanches and rock falls, and their simultaneous air blasts killed more than 350 people and destroyed the village of Langtang (Kargel et al., 2016). Satellite imagery and oblique helicopter photographs suggested that the deposit covering the village consisted of abundant snow and ice extending over an area of about 0.75 km$^2$. The extent of muddy deposits on the opposite slope (~200 m running up from the valley floor) provided an estimated debris speed of 63 m s$^{-1}$ at the valley floor. Based on a mean thickness ($\geq$2 m) and density (2200 kg m$^{-3}$) of the deposited material, the released gravitational potential energy was estimated as $\geq$3.2 $\times$ $10^{13}$ J, although the thickness was highly uncertain (Kargel et al., 2016). Utilizing a set of high resolution SPOT6/7 images, Lacroix (2016) estimated a deposit volume of $6.95 \times 10^6$ m$^3$ over the village, with a larger amount of avalanche material deposited above 5000 m a.s.l. ($9.66 \times 10^6$ m$^3$), and a total avalanche volume of $14.38 \times 10^6$ m$^3$, respectively. Differences



between two digital elevation models (DEMs) created from images on 21 April 2014 and 10 May 2015 showed distribution of not only the deposit but also destructed forest on the opposite slope (Fig. 5a in Lacroix, 2016) with errors not exceeding 3 m over the unaffected terrain. Nevertheless, the contributions and sequences of avalanches (snow and ice) and rock falls are not yet fully understood, though some of the event was described in Kargel et al. (2016).

In this study, we aim to unravel the sequence of the multiple events that led to this tragic event and quantify the contributions of snow, ice and rock by using in-situ observation from October 2015 in combination with pre-event data and aerial photography directly after the earthquake. We created DEMs and ortho-mosaiced images (ortho-mosaics) from oblique photographs taken by unmanned aerial vehicle (UAV) and multiple helicopter flights in May and June 2015 together with geographical information obtained by differential GPS (dGPS) survey. Using a satellite-based high resolution DEM as the

pre-event control, we estimate temporal changes in deposit volumes over the village. In addition, we investigated detailed structures of the deposit during the in-situ observation from which we estimate the volumetric contributions of snow avalanche, collapsed glacier ice and rock falls. During the in-situ survey and in Kathmandu, we interviewed villagers who witnessed the events in the valley and their narratives provide additional insights to the Langtang Village tragedy. By analyzing meteorological data recorded at multiple sites in the valley, we reveal that anomalous winter snowfall prior to the

earthquake might have amplified the co-seismic hazards.

## 2 Data and methods

### 2.1 Field observation

In order to survey the deposit covering the Langtang main village (Fig. 1), we carried out a field campaign in October 2015, during which we performed several flights with different UAVs in combination with dGPS surveys to create DEMs and

ortho-mosaics. We investigated the boundary between debris and ice, rock presence in the ice, and debris density in the ice at a few dozens of ice cliffs distributed near the Langtang River. We also observed size, condition, and age of trees flattened by the air blast on the opposite slope. During the in-situ observation and in Kathmandu, we interviewed 20 Langtang villagers to record their narratives of the events.

### 2.2 UAV campaign

We utilized a fixed wing UAV (Hobbyking SkyWalker X-5), which had a wing span of 1.18 m and a weight of 1.4 kg including camera. This plane is capable of approximately 45 minute flights at cruise speeds of 60 km h$^{-1}$ with either manual or autopilot modes. The camera mounted on the UAV, a Ricoh GR, has a lens of fixed focal length of 18.3 mm (equivalent to 28 mm focal length of 35 mm film camera) and 14.2 megapixel sensor, i.e. 4352 by 3264 pixels, capturing both JPEG and RAW format images in the visible light range. We performed two flights on 23 and 24 October 2015 for image acquisitions.

We launched the UAV from an upstream/eastern hill of the village (Fig. 2a). The desired image overlap was set to be greater than 70% in both lateral and longitudinal directions with respect to the UAV flight path. In order to generate the desired ground resolution finer than 0.1 m, the flight altitude was set 120-170 m from the ground.

### 2.3 dGPS survey and ground control points

In order to obtain precise positions of ground control points (GCPs) and validation data for generated DEMs, we conducted

dGPS surveys around the village using dual frequency carrier phase GPS receivers (GEM-1 and 2, GNSS Technologies, Inc., and R10, Nikon-Trimble Co., Ltd.). One receiver was set up at the Mundu campsite (Fig. 1) as a base station and two other receivers were operated in kinematic mode with a logging interval of 1-sec. The location of the base station was determined by the precise point positioning service operated by Natural Resources Canada (http://webapp.geod.nrcan.gc.ca/geod/tools-outils/ppp.php?locale=en, last accessed on 22 August 2016). We then post-processed the dGPS data with the RTKLIB



software (http://www.rtklib.com/, last accessed on 22 August 2016), from which the dGPS points calculated with the fixed solution were used in the following analyses. For the evaluation of DEMs generated from multiple sources, we convert the dGPS point data into gridded DEM of 1 or 5 m resolution (depending on the targeted DEMs with which the comparison made) by the inverse distance weighted (IDW) interpolation method with a search radius of $x/\sqrt{2}$ m (here x is the

resolution). Grid cells with no dGPS point were excluded following the method proposed by Fujita et al. (2009, 2011). Tshering and Fujita (2016) tested alternative interpolation methods such as IDW without fixing the search distance, arithmetic averaging and kriging with various settings and then demonstrated that the height difference from the IDW method (as root-mean-square errors) did not exceed 0.3 m except for the spline method.

During the survey, we evenly distributed and surveyed eight orange fabrics of 1 × 1 m over the deposit-covered area (Fig. 2).
We additionally surveyed seven obvious rocks located outside the deposit-covered area as the GCPs.

## 2.4 Generation of DEM and ortho-mosaic

We processed the UAV photographs (665 images, Table 1) into a DEM and ortho-mosaic using the Structure from Motion (SfM) workflow, which we implemented using the software package Agisoft PhotoScan Professional version 1.2.5 (Agisoft, 2016). Finally, we exported the DEM with a resolution of 0.2 m and the ortho-mosaic with a resolution of 0.06 m. For
estimating deposit volumes and comparisons with the dGPS data, we resampled the DEM to resolutions of 1 or 5 m depending on the targeted DEMs used for comparison.

## 2.5 Aerial photographs

As our UAV survey was conducted six months after the Gorkha earthquake, some of deposited snow and ice would likely have melted away. Some co-authors of this study have conducted aerial photography from a helicopter on the 7 and 10 of
May (D. F. Breashears) and on 1 June (H. Yagi). We processed these sets of photographs with the PhotoScan too. Unfortunately, no GPS measurements were acquired when the photographs were taken. In addition, the surface conditions drastically changed by the successive rock falls and the subsequent melting. For geo-referencing the imagery, we selected eight tie-points over an area unaffected by massive deposit (off-debris area hereafter) from the UAV-DEM/ortho-mosaic products (Fig. 2a). Details of camera and lens are listed in Tables 1. We processed these sets of helicopter photographs in
Agisoft PhotoScan with the same parameter settings as the UAV-DEM, and then generated three sets of helicopter based DEMs (heli-DEMs hereafter) and ortho-mosaics (7 May, 10 May and 1 June).

## 2.6 Pre-earthquake data

In order to estimate the deposit volumes and their changes over time, a pre-event DEM is required. We tested two DEMs (5-m resolution) derived from the 2.5-m resolution Panchromatic Remote-sensing Instrument for Stereo Mapping (PRISM) on-
board the Advanced Land Observing Satellite (ALOS). The first one is a DEM created with the ALOS-PRISM image acquired on 3 December 2010 (A10DEM). We first automatically produced an ALOS stereo-model and DEM with rational polynomial coefficients (RPCs) data on the Leica Photogrammetric Suite (LPS) software. Such an automatically extracted DEM often contains many errors especially in areas of highly irregular and abruptly changing topographies due to shadows and low contrast in the images. We therefore manually edited the point cloud by removal of false mass points in the air or
under the ground, and by placing adequate and representative mass points exactly on the terrain surface assisted by the LPS Terrain Editor (Sawagaki et al., 2012). The second DEM is a commercial product named 'ALOS World 3D' (AW3D), which is created by stacking and mosaicking using multi-temporal ALOS digital surface models to achieve the target accuracy within 5 m in both horizontal and vertical (Tadono et al., 2014).

In order to minimize horizontal and vertical errors, we compare both ALOS-DEMs of 5-m resolution with the UAV-DEM
over the off-debris area (Fig. 2). We horizontally shift the ALOS-DEMs at a 1-m step along south-north and west-east





directions, and then compared the result with the UAV-DEM resampled from 1-m resolution into 5-m resolution. The elevation difference and its standard deviation of the ALOS-DEMs against the UAV-DEM are depicted in Fig. 3. The performance of the AW3D (the minimum standard deviation of 1.46 m, also listed in Table 1) is much better than that of the A10DEM (the minimum standard deviation of 2.63 m). The DEM based on single scene (A10DEM) has large errors in the shaded portion along the Langtang River while stacking multiple images may reduce those errors (AW3D). Given this, we adopt the AW3D in the following analysis. Even in the AW3D, however, there are many void (no-data) areas at high elevation, where the glacier/snow surface lacks sufficient contrast to create stereo views. It is therefore difficult to evaluate the extent and amount of snow, ice and rocks that detached in the avalanche source area.

## 2.7 Automatic weather stations

To understand the interaction between meteorology, glaciology and hydrology, several automatic weather stations and precipitation gauges have been operated in the basin (e.g. Immerzeel et al., 2014; Ragettli et al., 2015; Shea et al., 2015) in addition to a long-term meteorological observation at Kyangjin, which was started during the mid-1980s (Fujita et al., 2006) (Fig. 1). The region is monsoon-dominated, and winter precipitation accounts for only 20-30% of annual totals (Immerzeel et al., 2014; Shea et al., 2015). In order to measure both liquid and solid precipitation precisely at high elevation, advanced pluviometers were installed near Yala and Gangja La Glaciers (YL_aws, YL_pv and GL_pv, respectively). Along a slope towards Gangja La Glacier, two more tipping buckets with temperature sensors (not air but instrument temperature) were installed (GL_tb1 and GL_tb2). Air temperature and snow depth were also measured at the three sites (YL_aws, YL_pv and GL_pv). All instruments, except for two tipping buckets, were damaged by a prior snow fall event or the air blast triggered by the Gorkha earthquake so that some parameters were not fully recorded. Precipitation, snow depth and air temperature data are retrieved from the data loggers and analyzed to understand the meteorological conditions until and during the event.

## 2.8 Recurrence period of winter snow

As described later, we find that the winter snowfall since November 2014 was anomalous. In order to evaluate how rare this winter snow event was, we calculate the recurrence period of the cumulative precipitation between 1 November to 25 April, which is a proxy for the total amount of snowfall. As the in-situ precipitation data observed at Kyangjin (Fig. 1) were available for only 12 years, it was not possible to calculate a probability density function. We therefore adopt the Aphrodite precipitation data for the period 1978-2007 (29 winters), in which the observed data was utilized (Yatagai et al., 2012) and thus representative even without calibration.

It has been reported previously that a vertical precipitation gradient exists along the slope between Kyangjin (3830 m a.s.l.) and Yala Glacier (5100 m a.s.l.) (Seko, 1987; Fujita et al., 1997; Immerzeel et al., 2014). As the station in Kyangjin malfunctioned in the winter of 2014-2015, we estimate cumulative winter precipitation from observations at Yala Glacier. Two precipitation gradients, 30.8% km$^{-1}$ (Immerzeel et al., 2014) and 38.0% km$^{-1}$ (Seko, 1987), which were observed precipitation ratios between Yala and Kyangjin in winter, were applied to the observations made near Yala Glacier.

To calculate the recurrence period, we tested major probability density functions such as Gumbel, generalized extreme value, lognormal, gamma, logistic and normal. Among all functions, which were acceptable by the Kolmogorov–Smirnov test, the Gumbel distribution function showed the best representativeness for the long-term gridded precipitation.

## 2.9 Interviews to local villagers

We interviewed 20 villagers in the Langtang valley and in Kathmandu during our visit from October to November 2015. We asked where the interviewee was during the earthquake and the amount of time between the earthquake and the first avalanche that hit the Langtang village. In addition, we asked about the color, wetness, and surface condition of the primary deposit. We exclude hearsay statements to avoid fictional narratives, which may have resulted from the traumatic course of



events. On the other hand, we corrected traditions with respect to the former huge earthquake in 1934 (Singh and Gupta, 1980) to understand how the village has relocated and expanded. All villagers verbally consented to the interviews.

## 3 Results

### 3.1 DEM evaluation

The dGPS survey method used in this study has a geodetic accuracy of ~0.2 m both horizontally and vertically (Fujita et al., 2008, 2009). By analyzing the differences between all DEMs with the dGPS data in the off-debris area (excluding built-up area, Fig. 2), we assess the accuracy of the DEMs (dz, Fig. 4 and Table 2). The mean deviations of the UAV-DEM (23 Oct) is ±0.26 m over the off-debris area and 0.34 m over the whole domain where elevation data is available. These errors are equivalent to or slightly greater than that from the dGPS method. Mean deviations of the other heli-DEMs over the off-

debris area increase toward 0.46 m (1 June), 0.96 m (10 May) and 1.51 m (7 May), respectively, while deviation of the AW3D (1.21 m) is of the same magnitude as those of two DEMs in May. In addition, deviations among the DEMs, which are necessary to evaluate uncertainty in volume change, are rather large ranging from 1.0 to 1.8 m (Table 2).

The heli-DEM on 7 May shows biases ranging from −1.37 to −1.00 m against the other heli-/UAV-DEMs (Table 2). Together with the positive bias of 1.54 m against the dGPS, this implies that the 7 May surface was higher than all other

dates (Fig. 4 and Table 2), and that the initial deposit was also present in the off-debris area. The spread of material outside the main debris deposit was also inferred from the boundary observed on the 7 May ortho-mosaic (Fig. 2d), although we selected obviously identifiable rocks as the tie-points on all ortho-mosaics. The AW3D DEM suffers from a significant negative bias of −5.08 m against the dGPS. For estimating deposited volume, the bias of AW3D is therefore corrected by increasing the elevation of the AW3D by 4.81 m, which is an average of biases of three DEMs (10 May, 1 June and 23

October) against the AW3D. Except for the AW3D and the heli-DEM of 7 May, biases among the other heli-/UAV-DEMs are within 0.4 m and therefore these were not corrected.

### 3.2 Change in deposit volume

Based on the multi-temporal DEMs, we evaluate thickness and volume changes of the deposit covering Langtang village (Fig. 5 and Table 3). The average thickness and total volume of the deposit by the primary avalanche, estimated from

elevation difference between the heli-DEM of 7 May and the AW3D over the deposit-covered area of 0.58 km$^2$, are 11.3 ± 1.8 m and 6.55 ± 1.07 ×10$^6$ m$^3$, respectively. Significant increases (>30 m) of surface elevation (blue color in Fig. 5a), suggesting deposition of avalanche material, are found along a gorge (locally named Pääbe Chu) situated at the western side of the village and in the Langtang River bed situated south of the village. A decrease of surface elevation (red color in Fig. 5a) on the opposite slope (southwestern part) is likely to be the results from destruction of forest by the air blast. If the

positive bias of the heli-DEM of 7 May found over the off-debris area (0.25 km$^2$, Figs. 2d and 5a) is taken into account as additional deposit, the total area and volume increase to 0.83 km$^2$ and 6.81 ± 1.54 ×10$^6$ m$^3$, respectively.

Between 8 and 10 May 2016, satellite images indicated a secondary large mass movement over the village (Kargel et al., 2016). Our heli-DEMs show that surface over the deposit-covered area elevated by 1.5 ± 1.6 m as an average thickness, corresponding to additional volume of 0.84 ± 0.92 × 10$^6$ m$^3$ (Fig. 5b and Table 3). In addition, Langtang villagers witnessed

rock fall on 12 May, which could have been triggered by the largest aftershock of M7.3 (Kargel et al., 2016) although the volume of this third large event could not be quantified.

During the three weeks between 7 May and 1 June, and the roughly five months between 1 June and 23 October, the surface elevation lowered on average by 2.0 ± 1.0 and 3.0 ± 0.7 m respectively, which is equivalent to volumes of 1.14 ± 0.58 and 1.64 ± 0.39 ×10$^6$ m$^3$, respectively (Figs. 5c and 5d, and Table 3). Ice covered by a thick debris mantle (> 0.5 m) is hard to

melt due to the insulating effect of debris (Mattson et al., 1993). The observed pattern of surface lowering suggests that ice





melting likely occurs at the bottom of the Pääbe Chu and of Langtang River (Figs. 5c and 5d), due to thermal erosion of the water flowing underneath.

### 3.3 Cheese boulder

Based on a suggestion and guided by two of the Langtang villagers, we surveyed a large boulder (Fig. 6), on which
sculptures of cheese and bread were carved in 1998 as the landmark for a cheese factory developed in the center of the village (hereafter referred to as the 'cheese boulder') (Fig. 6a). The cheese boulder was found on a terrace between the village and Langtang River (Fig. 6b). We surveyed both the original and found locations by the dGPS. The original location was difficult to identify as the scenery was completely changed by the deposit so we relied on guidance of the villagers. We took photographs of the boulder from multiple angles and positions, and then processed them in Agisoft PhotoScan to estimate
the volume of the boulder (Fig. 6c). We estimate density of the boulder by measuring weight and volume of ten samples of small stones of the same material as the cheese boulder. We calculated the volume of the cheese boulder to be $1.67 \pm 0.21$ m$^3$. With a measured density of $2.65 \times 10^3$ kg m$^{-3}$, a value of typical granite (Lillie, 1999), we determined the weight of the boulder to be $4.4 \pm 0.2 \times 10^3$ kg.

The villagers assert that the boulder was relocated by the blast because many farm animals, horses and yaks, were also
blown away. We therefore estimate initial speeds of the boulder at the original location by assuming three possible scenarios, which pass both original and found locations of the boulder: 1) no touchdown between both original and found locations (78 m s$^{-1}$, upper curve in Fig. 6d); 2) overshooting the neighboring 4.5-m height hill and then rolling down the slope (106 m s$^{-1}$, lower curve in Fig. 6d); 3) an eject angle of 45º giving the minimum speed (58 m s$^{-1}$, not shown in the figure). The estimated initial speeds are reasonably consistent with the speed of muddy flow (63 m s$^{-1}$), which was estimated from run up height on
the opposite slope (Kargel et al., 2016). This boulder movement can be utilized for validation of avalanche simulations in further study.

### 3.4 Other in-situ data

The extent of flattened forest on the opposite slope evidenced the remarkable impact of the air blast (Kargel et al. 2016). Field measurements of size, age and genera of fallen trees provide more insight into this event (Fig. 7a). The mean and
standard deviation of the diameter of 113 fallen trees are $0.16 \pm 0.07$ m (Fig. 8a). Most of them were uprooted because of thin soil layer on the bedrock. The age of trees, which consist of Abies and Rhododendron, is estimated to be around 35 years, which suggests that no event of this magnitude has occurred in the recent decades. Although a thirty-year mass balance record of Yala Glacier reconstructed from multiple sources suggested a large accumulation (solid precipitation) in 1975, 1976, 1980 and 1981 (Fujita et al., 2006), it is unknown whether the precipitation fell as winter snow, which could
have led to similar destructive avalanches, or as summer rain, which would not have affected the forest. There is no other record nor villagers' narrative suggesting any event to destruct trees in 1980 or earlier.

We performed visual investigations of the boundary between snow/ice avalanche and rock deposits, the thickness of debris on the avalanche deposit, and other features of the ice deposit. The boundary between ice and debris deposits at 20 exposed ice cliffs distributed near the Langtang River was clearly distinguishable (Fig. 7b), suggesting that the primary snow and ice
avalanche and subsequent rock falls occurred at different timings. Visual investigation of the ice tunnel and ice cliff over Langtang River also supports that these occurred as different events because no rocks greater than 1 m were found in the ice cliffs (Fig. 7c). Thickness of the debris mantle on the ice is $0.90 \pm 0.87$ m (Fig. 8b) however the thickness varies widely so that these would not be representative values due to a limited sample number and condition as of ice cliff exposure. In order to measure the density of sand contained in the ice deposit, we took six samples (weight ranging from 0.9 to 1.7 kg) from ice
cliffs. On average the percentage of sand and silt in the ice was $3.9 \pm 0.76\%$ in weight. Within the sand-rich avalanche deposit, we found rounded 'clear ice balls', whose diameter ranged in the order of several centimeters (Figs. 7d and 7e)



suggesting that this deposit could have different sources. We speculate that the dirty ice originated from entrained sand as a result of the snow avalanche, whereas the clear ice balls were derived from detached glacier ice that broke into small fragments without entraining dirty materials. Therefore, the ice cliffs we investigated were originally composed mostly of snow avalanche deposits, which was compressed, melted and refrozen to approximate ice.

## 3.5 Meteorological data

In mid-October of 2014 (14-15 October), a heavy snowfall event hit the valley during the passage of extratropical Cyclone Hudhud (Wang et al., 2015). Although this event was recorded in both the precipitation and snow depth records at all sites, the precipitation totals vary between sites while the snow depth data are more consistent at the different stations (Fig. 9a). Therefore, we calculate cumulative precipitation since 1 November by excluding records in October (Fig. 9b). Snow depth and cumulative precipitation show that, including Cyclone Hudhud, four major snowfall events occurred in mid-October, mid-December, early-January, and early-March, respectively (Figs. 9a and 9b). The instrument at Gangja La (GL_pv) stopped recording data on 2 January 2015, most likely because of an avalanche caused by a heavy snowfall event. However, this station supports the consistency of records among the meteorological sites and therefore it is included in the analysis. Precipitation records at the two Gangja La sites showed much less precipitation than those near Yala Glacier (Fig. 9b). This is partly because of the elevation gradient of precipitation (Immerzeel et al., 2014) but mainly because the tipping bucket tended to miss solid precipitation in winter. Comparing long-term averages of cumulative precipitation at Kyangjin (12 years, blue line with shading) and the Aphrodite precipitation data (29 years, purple line with shading, Yatagai et al., 2012), the amount of precipitation during the 2014/2015 winter was anomalously large.

Compared with the long-term average of air temperature at Kyangjin (3830 m a.s.l.), the air temperature recorded near Yala Glacier (4830 m a.s.l.) seems warmer than usual if a local and seasonal lapse rate of air temperature is taken into account (Immerzeel et al., 2014). However, the air temperature near Yala Glacier suggests that no snow melt occurred above 5000 m a.s.l. throughout the winter (Fig. 9c).

Temperatures measured inside the tipping bucket sensors decreased after the Gorkha earthquake and associated avalanche (Fig. 9d). Compared with two averages of diurnal change under normal conditions (blue lines with shadings), observed temperatures significantly dropped was synchronous with the timing of the earthquake and associated avalanche. The YL_pv record stopped because of the destruction due to another strong blast from a separate but coincident avalanche (orange line). It is unclear whether the observed temperature decreases were caused by air temperature cooling in the whole valley or by snow accretion to the instruments.

## 3.6 Recurrence period of the anomalous winter snow

We calculate winter snowfall recurrence periods to establish how rare the 2014-2015 winter snowfall was (Fig. 10). Although the recurrence period highly varies depending on various factors such as representativeness of the Aphrodite precipitation data and elevation gradient of precipitation, it suggests that the winter snowfall between November 2014 to 25 April 2015 was an extreme event which occurs only once during more than 100-500 years. Considering that the Aphrodite precipitation data is obviously greater than the observed one at Kyangjin (Fig. 9b), and that Cyclone Hudhud supplied more snow in October, the recurrence period of the anomalous winter snow could be greater than those evaluated above.

## 4 Discussion

### 4.1 Sequence of the events

Together with witness statements of villagers, we reconstruct the sequence of events triggered by the Gorkha earthquake as follows:


1) Triggered by the main shock of the Gorkha earthquake, a predominantly snow avalanche, with an estimated ice volume of $6.81 \times 10^6$ m$^3$, hit the Langtang village and covered the area of 0.58 km$^2$ with an average thickness of 11.3 m (Figs. 2d and 5a, Table 3). East and north of the main deposits, debris with an average thickness of 1.5 m (Fig. 5a) was spread over the village area of 0.25 km$^2$. Villagers witnessed that multiple avalanches hit the village within a short time frame and it is

difficult to attribute volumes to individual avalanches.

2) Glacier ice detached by the main shock could have triggered the avalanche though its amount would be limited to a few percent by volume (Sect. 3.4 and Fig. 7d). Visual investigation of high resolution Google Earth images confirms this. (Sect. 4.2)

3) The avalanche-deposited material also consisted of snow-entrained fine particle debris such as sand and silt, which is

estimated to be about 3-5% in weight density. The sand and silt were entrained during descent of the snow and ice over several thousands of meters (Sect. 3.4 and Fig. 7c) and most of villagers witnessed 'black avalanches'. Although some villagers observed rock fall after the first avalanche, witness statements about the slippery ice surface of the Langtang village and the photographs taken on 7 May (D. F. Breashears/GlacierWorks) indicate that rock fall was probably limited. This was already pointed out using the same oblique photographs and high resolution satellite images (Kargel et al., 2016).

4) Because of large elevation difference of 2500 m between source area (~5890 m a.s.l., Fig. 11) and the village (~3400 m a.s.l.), gravitational potential energy should melt snow and result in the 'wet avalanche deposit' that most of villagers witnessed. If all released energy was used to melt snow, water content is estimated to be 7.3% by following equation:

$$\theta = \frac{l_m}{gh} 100\% \qquad (1)$$

here $\theta$ is water content in weight (%), $l_m$ is latent heat for ice melt (333.5 J kg$^{-1}$), $g$ is gravitational acceleration (9.8 m s$^{-2}$), and $h$ is elevation difference (assumed to be 2500 m), respectively. It should be noted, however, that not all of the released gravitational potential energy would have been used to melt the deposited ice. Most energy would have been released in the air blast. Considering the volume of ice deposit ($6.81 \times 10^6$ m$^3$) and ice density of 900 kg m$^{-3}$, this further implies that the

energy of $1.5 \times 10^{14}$ J was released, which is 4.7 times greater than the previous estimate ($3.2 \times 10^{13}$ J by Kargel et al., 2016). 5) As Kargel et al. (2016) already pointed out, the released energy generated into a strong air blast, which swept away vulnerable nearby buildings and flattened mature forest on the opposite slope, which consisted of *Abies* and *Rhododendron* species with an average age of 35 years. Most of the trees were uprooted from the thin soil layer.

6) If the avalanche blew the cheese boulder away, the initial speed could have been 58 to 106 m s$^{-1}$, depending on a number

of assumptions (Fig. 5d). This estimate is reasonably consistent with the other estimation of the minimum speed of the air blast at the valley bottom (63 m s$^{-1}$) based on the run up height of muddy material on the opposite slope (Kargel et al., 2016)

7) Some villagers witnessed that the deposit covering the village also temporarily dammed the Langtang River but an ice tunnel formed quickly and the river water drained without causing an outburst flood.

8) Between 8 and 10 May, co-seismic rock falls occurred and the surface covering the Langtang village changed from ice

and snow to rock debris. The event resulted in approximately $0.84 \times 10^6$ m$^3$ of debris and rocks and elevated the deposit surface by 1.45 m in average (Figs. 2c and 5b, Table 3). This is also evidenced by the photographs taken on 10 May (D. F. Breashears/GlacierWorks) as Kargel et al. (2016) described.

9) A villager who has been searching for the missing villagers witnessed that additional rock falls, which were triggered by the largest aftershock on 12 May, hit and covered the village. However, its thickness and volume are not quantifiable.

10) During the succeeding three weeks between 10 May and 1 June, the surface elevation lowered by 2.0 m suggesting that the $1.14 \times 10^6$ m$^3$ of ice was melted away (Figs. 2b and 4c, Table 3). If the additional rock debris deposited on 12 May is taken into account, by assuming average thickness of 1 to 2 m, the melted ice could be 1.5 to 2.0 times greater than the above estimation. During the roughly five months including monsoon season, the surface elevation lowered by a further 3.0





m, suggesting that the $1.64 \times 10^6$ m$^3$ of ice was melted away (Figs. 2a and 5d, Table 3). The greatest surface lowering occurred along the Pääbe Chu between Gomba and the village, and along Langtang River (Figs. 5c and 5d). Because ice covered with thick debris (> 0.5 m) melts slowly due to the insulation effect of the debris mantle (Mattson et al., 1993), the high melt rates are probably caused by thermal erosion of the water flowing underneath. The debris-covered area was

slightly reduced to 0.54 km$^2$ during the monsoon.

11) By the end of October 2015, six months after the main shock, about 60% of the initial volume of deposited ice and snow remained over the village.

### 4.2 Avalanche sources

A set of pre- and post-event images of SPOT6/7 successfully quantified the mass movement in the Langtang valley due to

the earthquake-induced avalanche (Lacroix, 2016). The main deposit covering the Langtang village was estimated to be 6.95 $\times 10^6$ m$^3$, which is consistent with the volume evaluated in this study (6.81 $\times 10^6$ m$^3$). At high elevations the imagery was of high quality (clear and with a lot of contrast) (Fig. 5b in Lacroix, 2016) such that the estimated volume (14.38 $\times 10^6$ m$^3$), which was detached from Mt. Langtang Lirung, seemed to be reliable. It is problematic, however, that the pre-event image was taken on 21 April 2014, one year earlier than the event, so that snowfall on and melting of the glacier surface during the

2014 monsoon season were fully ignored in the differentiation of two DEMs. Given the fact that the amount of snow prior to the event was anomalous, it is likely that the detached volume should have been greater than 14.38 $\times 10^6$ m$^3$. Another potential source of error are the glacier dynamics, by which the glacier surface tends to be lowered at high accumulation zone (Cuffey and Paterson, 2010). In addition, change in density from snow to ice was not taken into account. Although other images available on Google Earth were carefully checked to confirm that no landslides had occurred during the 2014

monsoon season, none of the above issues were addressed in the source region of the main avalanche.

Visual inspection of the pre- and post-event satellite images on Google Earth revealed a collapse of hanging glacier ice (Fig. 12). We roughly estimate volume of this ice block to be $66 \times 10^3$ m$^3$ by assuming thickness of the glacier ice thickness as 30 m, which is a similar value of Yala Glacier (Sugiyama et al., 2013), and this accounts for only 1% of the initial avalanche deposited. Other possible areas for the avalanche source are highly distorted because of insufficient ortho-rectification of the

post-event image (3 May 2015) which complicates the evaluation of the contributing volume of the collapsed glacier ice. Together with the in-situ observation of clear ice balls in the dirty ice deposit (Fig. 7d), which likely originated from glacier ice, we assume that the contribution of collapsed glacier ice could be a few percent. However, it should have played a key role in initiating the entire event.

Meteorological data suggest that multiple snowfall events have accumulated seasonal snow with a thickness of up to 1.5 m

above 5000 m a.s.l. during the 2014/2015 winter (Fig. 9a). Limiting the source area of the avalanche to areas higher than 5000 m a.s.l. (8.73 km$^2$, Fig. 11), above which no snow melt can be assumed in winter (Fig. 9c), the deposit over the Langtang village (6.81 $\pm$ 1.54 $\times 10^6$ m$^3$) is equivalent to 0.78 $\pm$ 0.18 m of ice thickness over the entire the source area. Assuming that the settled snow density of the winter snow is 450 kg m$^{-3}$ (half of the ice density of 900 kg m–3), the snow thickness and volume over the avalanche source area are estimated to be 1.56 $\pm$ 0.35 m and 13.62 $\pm$ 3.08 $\times 10^6$ m$^3$,

respectively. This snow thickness is surprisingly consistent with the observed snow depths near Yala Glacier (1.28 m at 4830 m and 1.52 m at 5100 m, Fig. 9a). Although the volume detached from the upper catchment (14.38 $\times 10^6$ m$^3$) estimated by Lacroix (2016) seems consistent with the estimate in this study, this could be a mere coincidence if both gain and loss of mass on a typical Himalayan glacier in summer season (Ageta and Higuchi, 1984), the glacier dynamics (Cuffey and Paterson, 2010), and the winter snow are taken into account.



### 4.3 Anomalous winter snow

Although there is no doubt that the 2015 Gorkha earthquake triggered the primary snow avalanche that destroyed the Langtang village, the anomalous winter snow, which was an extreme event having the recurrence period of 100 to 500 years, has likely amplified the disaster that took place. By villagers statements, coincident with the previous large earthquake (M8.3) recorded in 1934 (Singh and Gupta, 1980), ice avalanches hit the Numthang and Mundu areas, which were the former main village (Fig. 1). It is unfortunate that, since then, the village has expanded towards the present village location, and that the trekking boom has accelerated this expansion since the 1990s. It is evident that the unlikely combination of an anomalous amount of snow and earthquake with this magnitude have contributed to the magnitude of this disaster. It is still unclear whether such an amount of anomalous winter snow could cause such a huge avalanche by itself without an earthquake trigger. Nor is it clear why similar huge avalanches have not been initiated during the earthquake in other neighboring districts in Nepal.

### 5 Conclusions

We use multiple lines of evidence to re-construct the series of events leading to the devastating avalanche that occurred in Langtang Valley, Nepal, during the Gorkha earthquake. Through intensive in-situ observation in October 2015, we evaluate the volume and structure of the deposit covering the Langtang village, and sequence of the multiple events. Multi-temporal digital elevation models created from photographs taken by helicopter and unmanned aerial vehicle together with differential GPS survey reveal that the deposit volumes of primary and succeeding events were $6.81 \times 10^6$ m$^3$ and $0.84 \times 10^6$ m$^3$, respectively. Visual investigations around ice cliffs exposed around Langtang River and witness statements of villagers suggest that the primary event was snow avalanche while the contribution of the collapsed glacier ice could account for only a few percent. Our observations also indicate that successive rock falls, possibly triggered by aftershocks up to nearly three weeks after the initial earthquake, buried the initial deposit.

An average snow depth (1.56 m) estimated from (a) the deposited volume, (b) the upper catchment area above 5000 m a.s.l., and (c) density assumptions of snow and ice is surprisingly consistent with snow depths (1.28-1.52 m) observed at a nearby meteorological stations (4830-5100 m a.s.l.). Both amounts are anomalously large, as a result of four major snowfall events that occurred since October 2014. Considering the vertical gradients of winter precipitation (e.g. Collier and Immerzeel, 2015), the 1300 m difference in elevation between the lower village (where long-term observational data is available) and the high elevation stations, and a range of probability density functions, we conclude that this anomalous winter snow was an extreme event, which occurs only once every 100 to 500 years. The anomalous snow conditions likely amplified the most remarkable disaster induced by the 2015 Gorkha earthquake in Nepal. To support the village and community reconstructions in Langtang Valley, an avalanche hazard map based on the facts revealed through this study will be provided to the villagers' community and local authorities.

**Author contributions.** K.F. designed the study. H.I., T.I, W.W.I. and J.M.S conducted the UAV operations. K.F., S.Y., S.S., K.N. and H.Y. conducted the field investigations. A.Sadakane, S.Y. H.I. and K.Y. interviewed the villagers. R.B.K. supported the field investigations. D.F.B. and H.Y. provided the helicopter oblique photographs. A.Sakai, T.S. and S.S. analysed the ALOS satellite data. K.F. wrote the paper. W.W.I. and J.M.S contributed to the discussion.

**Acknowledgements.** We thank the Department of Hydrology and Meteorology, Nepal, for providing the opportunity and permission to conduct the field observations, and access to the past meteorological data at Kyangjin. We are indebted to Guide For All Seasons for its logistical support, and to Prodrone Co. Ltd. for its dedicated support on UAVs. We acknowledge G. Silwal, H. Watanabe, R. Dahal and N. Thapa for their support in the field. Thank also goes to M. Sano for his support in detecting tree genera and counting tree ring. We thank K. Arita and J. Maeda for their suggestion on granite density. This study was supported by J-RAPID of Japan Science and Technology Agency, JSPS-KAKENHI grant Number





15H05793, and the Inoue Fund for Field Science of the Japanese Society of Snow and Ice. The work of W. Immerzeel was supported by funding from the European Research Council (ERC) under the European Union's Horizon 2020 research and innovation program (grant agreement No. 676819). J. Shea was supported by ICIMOD through the Norwegian-funded Cryosphere Initiative. ICIMOD is funded in part by the governments of Afghanistan, Bangladesh, Bhutan, China, India,
Myanmar, Nepal, and Pakistan. The views expressed are those of the authors and do not necessarily reflect their organizations or funding institutions.

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



Table 1. Camera and lens properties of UAV and helicopter photographs and resolution of the SfM products.

| Date | Camera | Focal length* | Pixels | n+ | Resolutions (m) # | Photographer |
|---|---|---|---|---|---|---|
| 7 May | Nikon D800E | 24 mm | 7360 by 4912 | 69 | 0.29 / 0.30 | D. F. Breashears |
| 10 May | Nikon D800E | 24 mm | 7360 by 4912 | 92 | 0.06 / 0.10 | D. F. Breashears |
| 1 June | Canon EOS 5D | 28 mm | 4368 by 2912 | 322 | 0.65 / 0.67 | K. Yagi |
| 23 October | Ricoh GR | 28 mm | 4352 by 3264 | 665 | 0.06 / 0.06 | H. Inoue as UAV operator |

* as of 35-mm camera

+ number of photographs used in the SfM analysis

5    # Resolutions of digital elevation model and ortho-mosaic




Table 2. Accuracy evaluation of digital elevation models over off-debris area (Figs. 3 and 5). Average and standard deviation of elevation difference (unit is meter) are listed at below and above the diagonal, respectively. Subtraction is performed as recent DEM (lower in the column or righter in the line) is subtracted by previous one. Cell size for the comparison with AW3D is 5-m while the others are 1-m. Comparison with dGPS data is shown in Fig. 4.

| | dGPS | AW3D | 7 May | 10 May | 1 Jun | 23 Oct |
|---|---|---|---|---|---|---|
| dGPS | | 1.21 | 1.51 | 0.96 | 0.46 | 0.26 |
| AW3D | −5.08 | | 1.85 | 1.81 | 1.31 | 1.46 |
| 7 May | 1.54 | −3.60 | | 1.58 | 1.25 | 1.36 |
| 10 May | 0.24 | −4.86 | −1.26 | | 1.02 | 1.21 |
| 1 Jun | −0.01 | −4.97 | −1.37 | −0.11 | | 0.72 |
| 23 Oct | 0.24 | −4.60 | −1.00 | 0.26 | 0.37 | |



Table 3. Changes in mean thickness (below the diagonal, m) and volume (above the diagonal, million m$^3$) of the debris covering the Langtang village. The standard deviations obtained over the off-debris area (Table 2) are used to estimate the error in volume.

| | AW3D | 7 May | 10 May | 1 Jun | 23 Oct |
|---|---|---|---|---|---|
| AW3D | | 6.55 ± 1.07 | 7.39 ± 1.05 | 6.34 ± 0.74 | 4.69 ± 0.78 |
| 7 May | 11.31 ±1.85 | | 0.84 ± 0.92 | −0.31 ± 0.72 | −1.93 ±0.76 |
| 10 May | 12.76 ±1.81 | 1.45 ±1.58 | | −1.14 ± 0.58 | −2.73 ± 0.68 |
| 1 Jun | 11.27 ±1.31 | −0.55 ±1.25 | −2.00 ±1.02 | | −1.64 ± 0.39 |
| 23 Oct | 8.75 ±1.46 | −3.46 ±1.36 | −4.90 ±1.21 | −2.98 ±0.72 | |





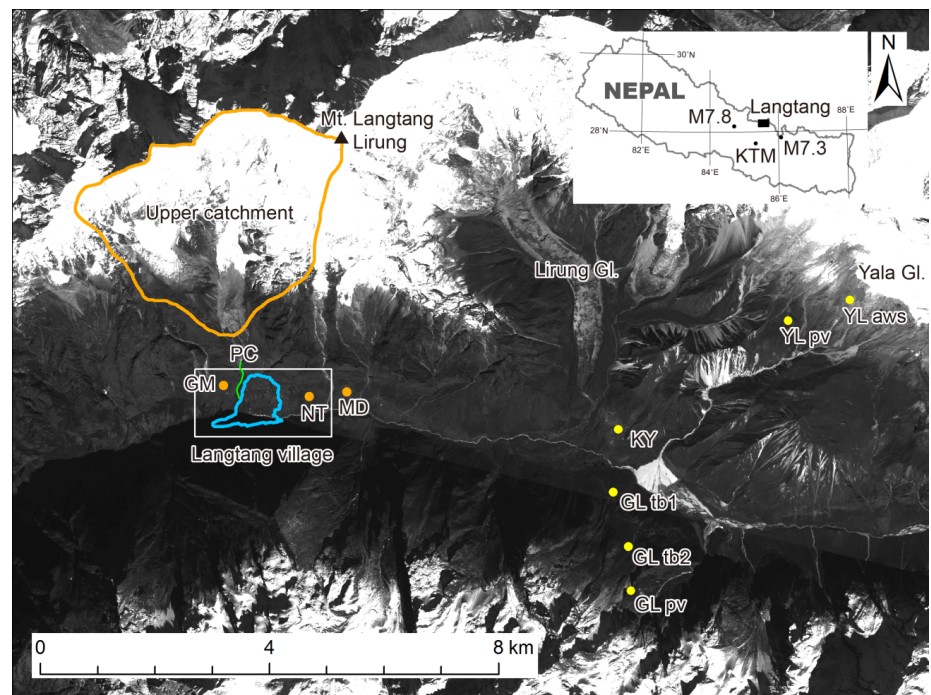

**Figure 1: Langtang village surveyed in this study (white polygon indicating area of Figs. 2 and 5), debris covering the village (blue polygon), upper catchment from which the avalanche could have originated (orange polygon), other villages mentioned in the text (orange circles), small gorge (green line), and locations where meteorological stations are located (yellow circles) in the valley (valley location shown in inset panel). The background is based on an ALOS-PRISM image taken on 3 December 2010 before the Gorkha earthquake. Abbreviations denote GM: Gomba, PC: Pääbe Chu, NT: Numthang, MD: Mundu, KY: Kyanjin, YL: Yala, GL: Gangja La, aws: automatic weather station, pv: pluviometer, and tb: rain gauge of tipping bucket.**





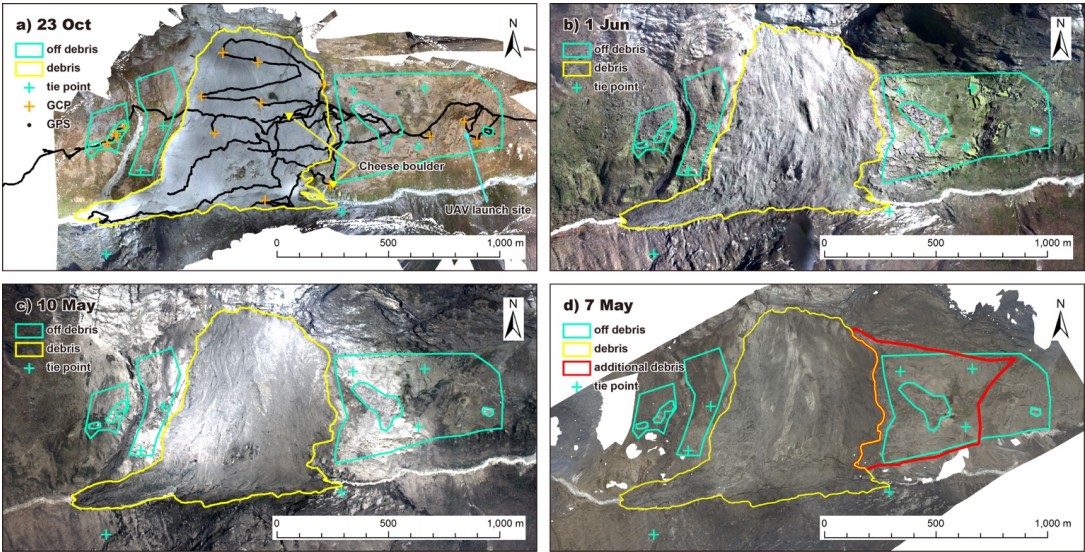

Figure 2: Orthoimages over the Langtang village on (a) 23 October, (b) 1 June, (c) 10 May and (d) 7 May of 2015.





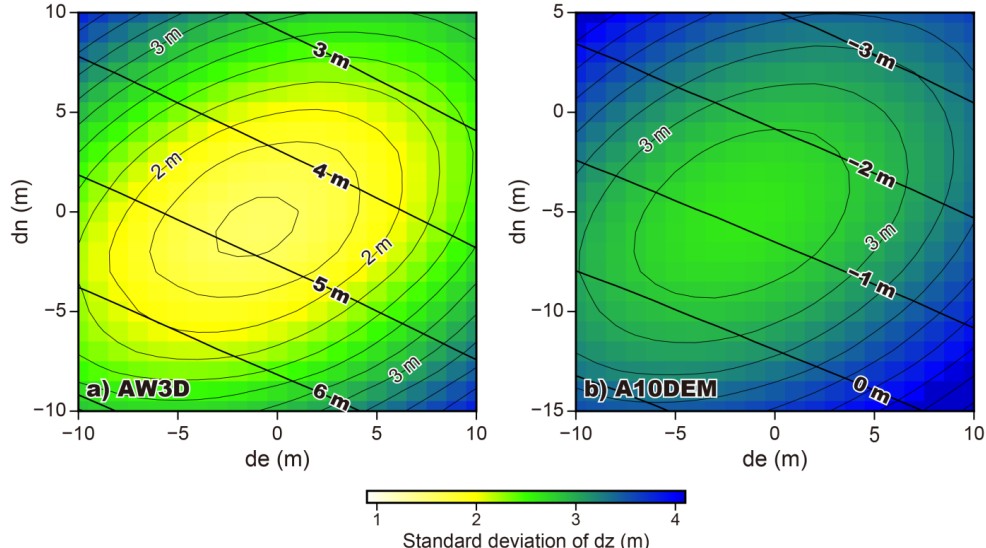

**Figure 3: Distribution of elevation bias (dz, thick contour lines) and its standard deviation (color shadings with thin contour lines) of (a) AW3D and (b) A10DEM against the UAV-DEM on 23 October, which are calculated over the off-debris area (Fig. 2a). ALOS-DEMs are shifted at 1-m interval to both directions of west to east (de, horizontal axis) and of south to north (dn, vertical**
5 **axis), and then mean elevations for 5-m cells (corresponding to the ALOS-DEMs) are calculated from the UAV-DEM, which is resampled to 5-m from the original 1-m resolution.**


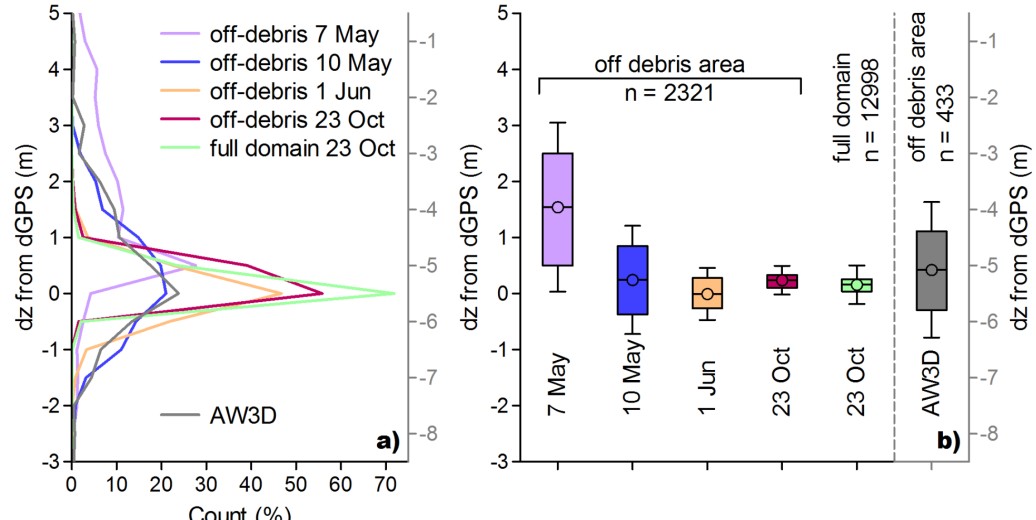

**Figure 4: Evaluation of DEMs, expressed as (a) histogram and (b) box plot of elevation difference from dGPS data (dz, vertical axes). Evaluation was performed over the off-debris area (Fig. 2a), at which no/less debris was found, while 'full domain' denotes the whole area at which both elevation data of UAV-DEM and dGPS were obtained during our in-situ observation in October 2015. The cell size of the DEMs is 1-m except for that of the AW3D (5-m). Also note the bias in the AW3D (left axes). Box, line and circle in box and whisker denote interquartile, median and mean, and one standard deviation, respectively.**





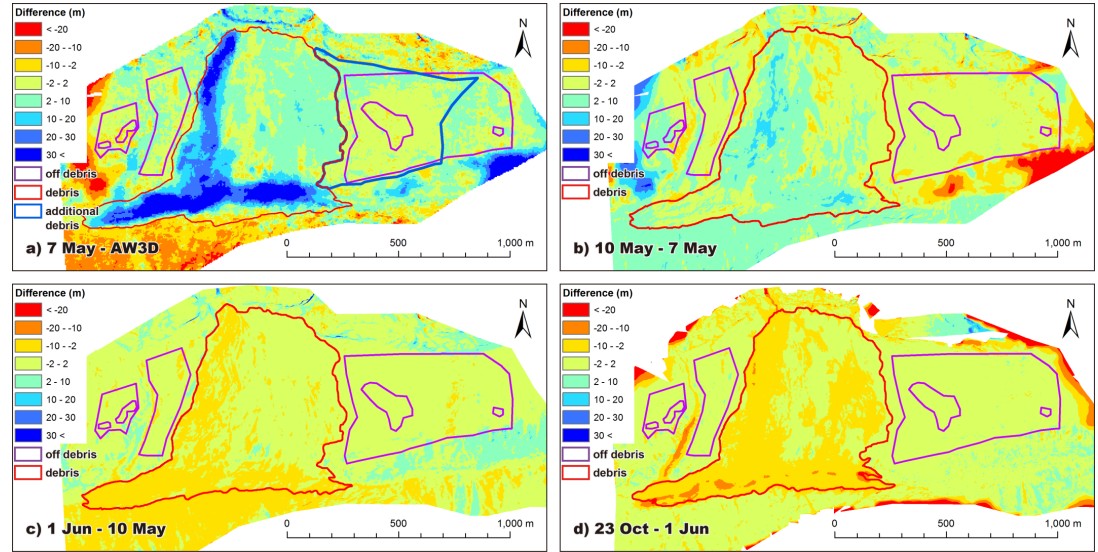

**Figure 5: Elevation difference over the Langtang village between (a) 7 May 2015 and AW3D, (b) 10 and 7 May, (c) 1 June and 10 May, and (d) 23 October and 1 June. The difference is obtained by subtracting the pre-DEM from the post-DEM.**





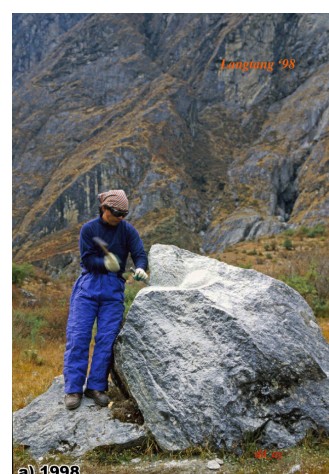
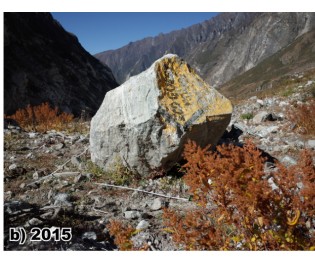
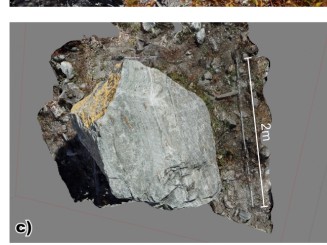
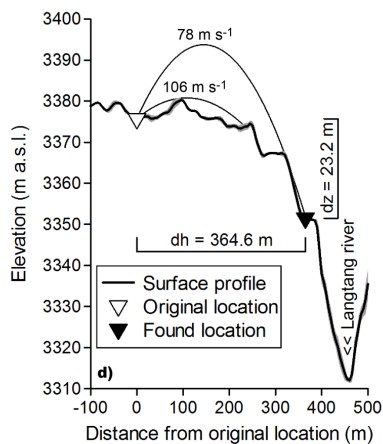

**Figure 6: Photographs of the 'cheese boulder', (a) when the sculpture was carved in 1998, (b) during our in-situ observation (22 October 2015), and (c) screenshot from the structure from motion analysis with the similar angle of (a). (d) Cross section profile along which the boulder moved. Locations (Fig. 2a) and the surface profile were obtained by dGPS survey and the UAV-SfM products. Elevation error is calculated based on the UAV-DEM (1-m resolution) by computing a standard deviation within a 5 m distance. Thin parabolic curves denote possible passes, along which the boulder was blown. Higher curve does not touch the ground between the original and found locations while lower curve can cross the 4.5-m height hill, and then the boulder could roll down. Also shown are the initial speeds at the original location for the boulder to have the passes.**





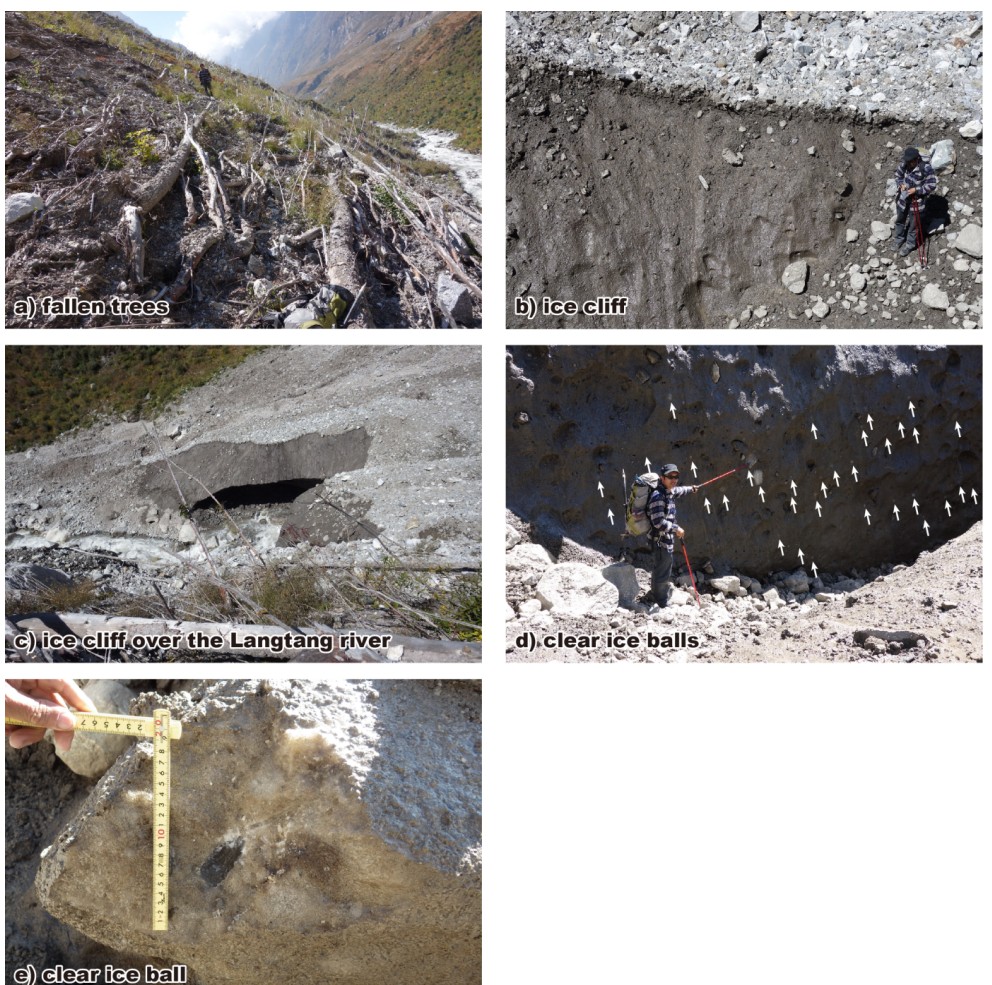

**Figure 7:** In-situ photographs of (a) fallen trees, (b) boundary between ice and debris mantle at ice cliff, (c) rock existence in ice cliff over Langtang River, (d) 'clear ice balls' contained in 'dirty ice' pointed by white arrows, and (e) a close-up of clear ice ball.





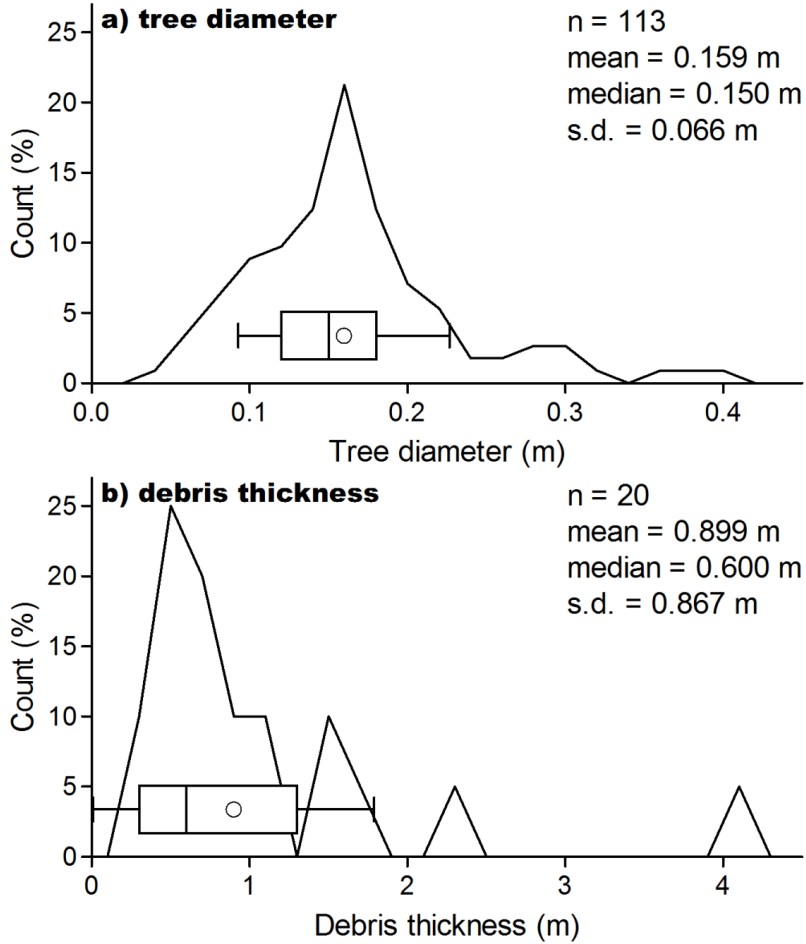

**Figure 8:** Histogram and box plot of (a) diameter of trees fallen on the opposite slope of the Langtang village and (b) debris thickness found at ice cliffs. Box, line and circle in box and whisker denote interquartile, median and mean, and one standard deviation, respectively.





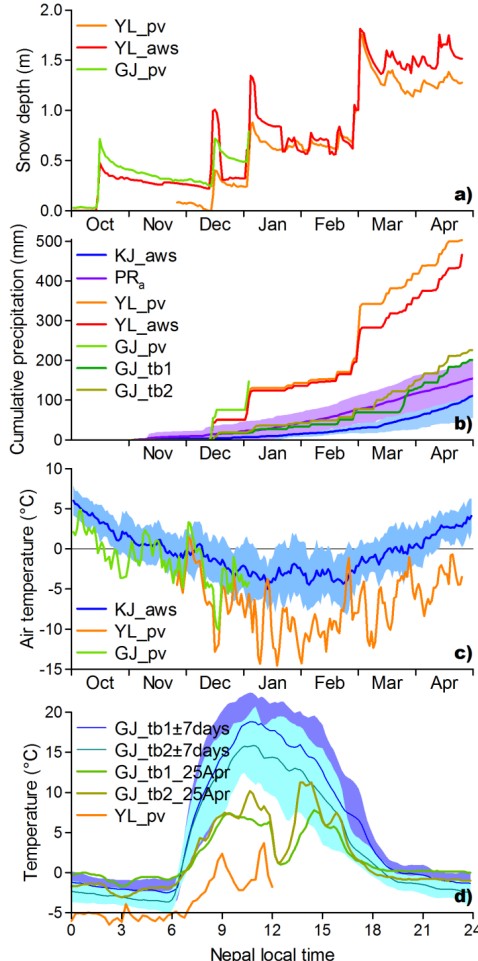

Figure 9: (a) Snow depth, (b) cumulative precipitation, and (c) air temperature observed at meteorological sites in Langtang Valley until the Gorkha earthquake (25 Apr. 2015). Blue lines with shadings in (a) and (c) denote long-term means observed at Kyangjin (KJ_aws, 1988-1992, 2002-2008, 12 years in total). Purple line with shading in (a) denotes long-term means of Aphrodite precipitation at a grind including Kyangjin (PRa, 1978-2007, 29 years in total). (d) Diurnal temperatures recorded by a sensor embed in tipping bucket at two Gangja La sites. Thick green lines and thin blue lines with shadings denote temperatures on 25 April 2015, the date when the Gorkha earthquake occurred, and means created from pre-/post-weeks (14 days in total but excluding 25 April) of the earthquake. Also shown is air temperature recorded at Yala site (orange line), which was destroyed by wind blast. Locations of the site are shown with abbreviations in Fig. 1.




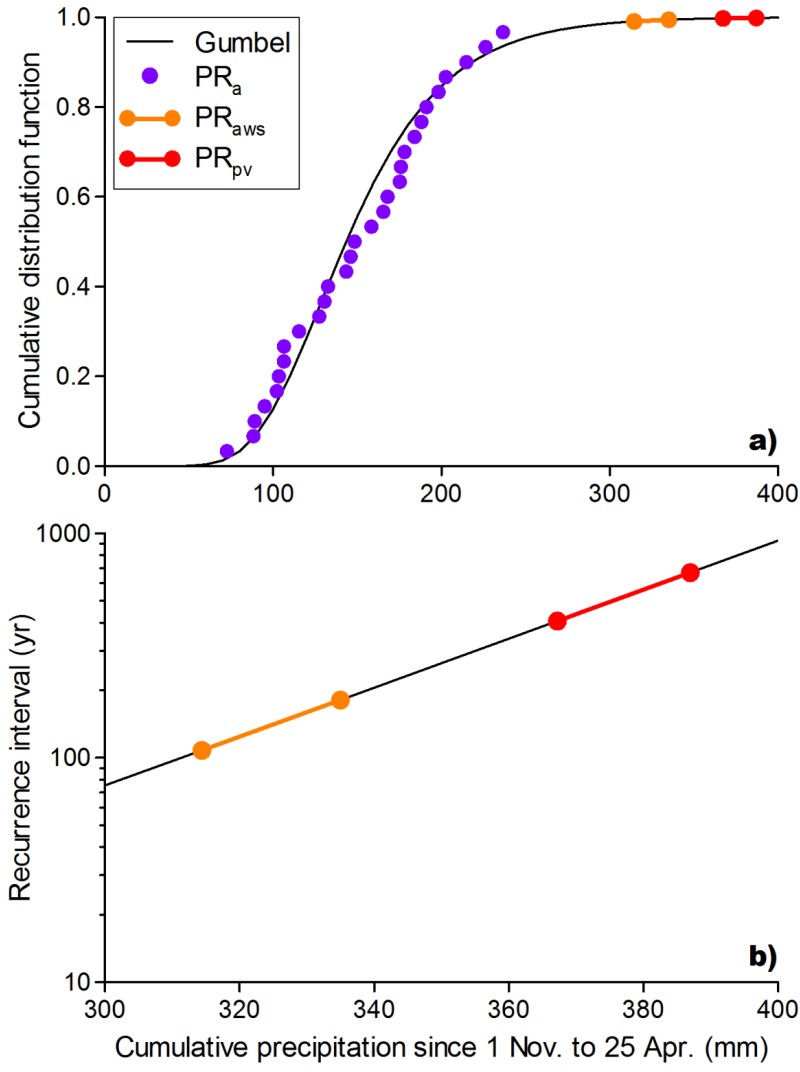

**Figure 10: (a) Cumulative density function of cumulative precipitation since 1 November to 25 April (PR$_a$, purple circles) and the Gumbel density functions (black line), and (b) recurrence interval. Red and orange circles denote calibrated precipitation for the elevation of Kyangjin (3830 m a.s.l.) from two sites near Yala Glacier (Fig. 1) assuming two elevation gradients for precipitation.**



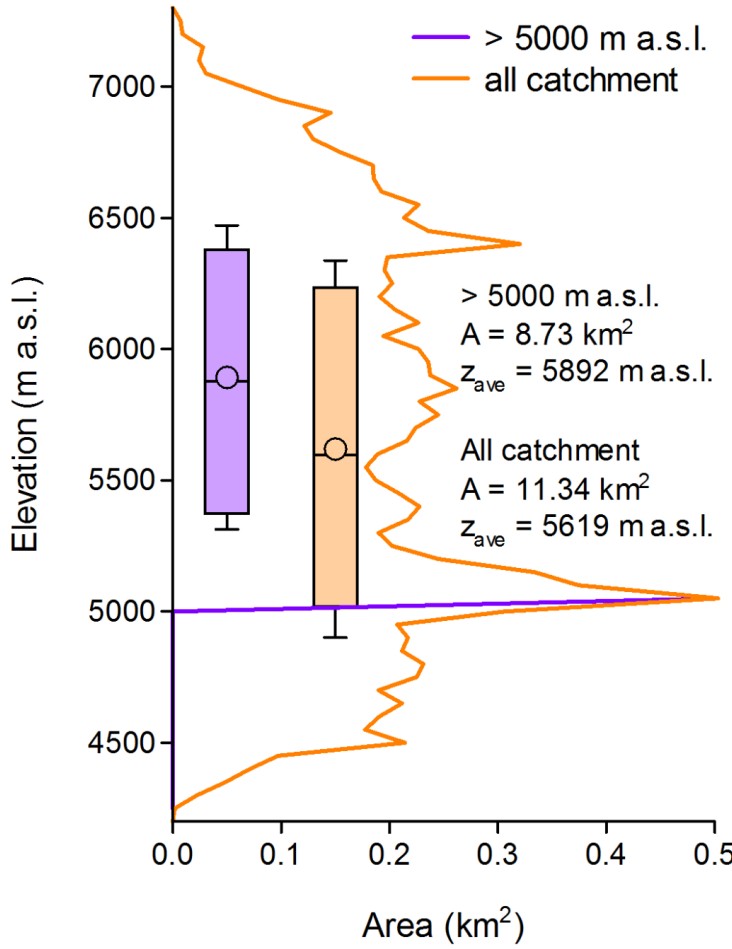

Figure 11: Hypsometry and box plot for the upper catchment above the Langtang village, from which the main avalanche might have occurred (orange polygon in Fig. 1). Also shown is that limited above 5000 m a.s.l. Elevation data is used from ASTER-GDEM2 (Tachikawa et al., 2011). Box, line and circle in box and whisker denote interquartile, median and mean, and one standard deviation, respectively.





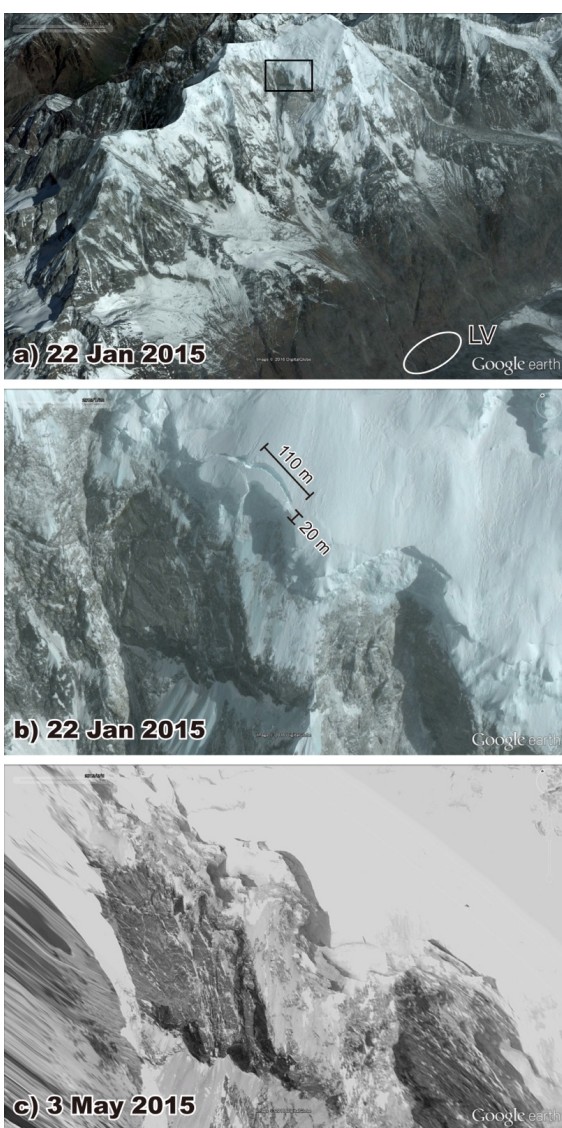

**Figure 12: An example of collapsed glacier ice captured from the Google Earth. Panels are a) overview of the southwest face of Mt. Langtang Lirung, b) ice block found in the pre-event image on 22 January 2015, and c) the ice lost from the post-event image on 3 May 2015, respectively. Black box and ellipse in panel (a) denote area of panels (b) and (c), and the damaged Langtang village (LV), respectively.**