# Peer review of "Anomalous winter snow amplified earthquake induced disaster of the 2015 Langtang avalanche in Nepal"

_Natural Hazards and Earth System Sciences, 2016_

## Referee Comment (RC1) · Anonymous Referee #1 · 3 Nov 2016

The authors present interesting data and hypothesis on the disastrous ice avalanche event triggered by the Gorkah earthquake in the Langtang valley.

However the paper mixes up facts measured by different methods (weather stations, photogrammetric DSMs, statistics) with quite bold hypothesis on the process which are not underlay sufficiently by facts and measurements. The authors have to state clearly which stamens are based on facts and which statements are hypothesis or guesses. Right now these two types of stamens are dangerously mixed up.

I do not agree on the hypothesis of the avalanche with only marginal contribution of glacier ice. The letter by Lacroix (2016), which is cited by the authors, clearly shows that considerable mass of up to 30 m thickness detached close to the ridge at several locations. This must be glacier ice and cannot be snow (estimated snow thickness 1.5

m). So the major part of the mass hate to come from glacier ice. The whole part on the triggering and the dynamics of the event is very weak. Based on the presented data no sound standing explanation of the triggering and dynamics of the event is possible. The authors should therefore delete or at least substantially reduce this part and declare it as hypothesis. As far as I know David Breashears also scoured high resolution photographs of the upper part of the area. Why are these photographs not used to generate a DSM?

Also the story about the Chees Boulder is very vague. I agree that it is very interesting but I do not think that the facts are sound enough to include it into a scientific publication.

In my opinion parts of the presented data is interesting. However in the presented form the paper is not acceptable for publication in a scientific journal and has to be carefully overworked by clearly dividing measured facts and reducing speculations not supported by sufficient measurements or observations.

---

## Short Comment (SC1) · 17 Jan 2017

The submitted manuscript by Fujita et al. documents the extent of the earthquake-induced Alpine mass motion (snow/ice avalanches and rock falls), which destroyed and covered the village of Langtang (Nepal) as a result of the 2015 Gorkha earthquake. Combining differential GPS measurements after the rock fall/avalanche events with digital elevation models derived from satellite and airborne (helicopter and UAV) measurements, the authors estimate a volume of the debris cover and its evolution over time. This constitutes one of the two central messages, which this study provides. For the second message, the authors use meteorological data and further imagery to suggest source constraints, namely that the primary avalanche damage was triggered by glacier collapse and unusual pre-event winter precipitation magnified the extent of the destruction.

The paper is clearly written and presents a valuable assessment of a tragic natural disaster. The provided geographic constraint on debris cover seems somewhat detached from the discussion of the avalanche source. However, since the Gorkha earthquake and its damage are still relatively recent events, it should be OK to present such different aspects in a single paper.

My main criticism deals with several assumptions, which the authors seem to make but do not fully justify. Most of all, did the increased pre-event winter snowfall really make a significant difference? On Page 4, Line 13 the authors state that for the region of concern, only 20-30% of the annual totals in snow accumulation are attributed to winter snow fall. Does this mean that similar or much larger avalanches are typically to be expected after the monsoon season? Do the authors assume that winter snow cover is particularly vulnerable to earthquake shaking?

Additional explanations and/or specifications of assumption are necessary for the subtracting of DEM's, the discussion of the hanging glacier collapse and calculation of avalanche source volume. Revisions should not require additional analysis. However, in its current state, the manuscript is not entirely clear on how the conclusions are reached. Result presentation and figure quality could also be improved, but these are likely more minor points.

MAIN REMARKS

Extreme snow fall: As mentioned above, it is not clear why the winter snowfall, even though it was particularly strong, amplified the avalanche devastation significantly. This seems to imply that either snow avalanche risk in in spring 2015 was larger than during the rest of the year and/or the earthquake was more likely to happen after winter than after summer (when most of the snowfall occurs). Neither of these points is argued for. As a first step, the time series in Figures 9a-c should be extended to a full year to show if solid winter precipitation dominated solid summer precipitation. Next, a discussion about earthquake-triggered avalanches would be helpful. Podolskiy et al. (2010 in

Journal of Glaciology) conclude that during earthquake shaking, avalanche failure occurs primarily along weak layers or the base of snow samples. Is such triggering more likely during certain seasons? Alternatively, if the trigger was entirely due to glacier collapse, is this more likely during winter? Finally, it would help to see an estimate of how much avalanche-prone snow volume is usually available for these kinds of events, excluding extreme snowfalls such as documented in the manuscript.

DEM Generation: This discussion inevitably involves lots of numbers. Unfortunately, the authors sometimes round numbers, and sometimes they do not. Also, standard deviation is not always indicated. This makes it very hard and tedious for the reader to flip between Tables 2,3 and the text. I suggest more consistency. Furthermore, I could not follow the reasoning behind the "initial deposit" (Line 15 on Page 5). I thought the off-debris area was deposit-free? I guess that fresh debris deposits would have been noticed during the GPS survey. Is it reasonable to believe the large bias given the similarly large standard deviation? I am not an expert on DEM comparison, but I expect that some statistical argument is needed here.

Projectile motion of boulder: I did not understand how the initial launch speed of the boulder relates to the avalanche. The authors seem to argue that the air blast (I suggest clearly defining this term) ahead of the avalanche launched the boulder. However, on Line 19 of Page 6 they compare boulder speed with "speed of muddy flow". In any case, there should be some explanation on moment transfer between avalanche debris or air and boulder. Is there really a simple explanation why the different speeds should be the same?

Glacier trigger: The authors provide evidence for a glacier collapse that could have triggered the avalanche. They should specify if this is the only collapse that occurred in the time window of interest or not. In addition, one of the conclusions of this study seems to be that glacier ice did not exceed a few percent of the avalanche volume. In contrast, on Line 38 of Page 6 they say that their ice cliff samples may not be representative. Generally, I suggest being clearer on the glacier collapse trigger, because

this is a poorly understood topic in glaciology. To my understanding, it is not generally clear how glacier and underlying bedrock failure play a part in large-scale avalanches.

Presentation of geographic and timing information: The manuscript would benefit from more altitude information. All maps/photos should include altitudes of mountain peaks, villages and perhaps other sites such as glacier tongues. Currently, it is hard to put the glacier collapse in context without knowing its altitude with respect to the rest of the avalanche source area. Moreover, what are the approximate glacier volumes? In section 4.2, the authors state that the collapsed glacier cliff has a similar volume to all of Yala Glacier. This seems very small. I furthermore suggest specifying the earthquake date whenever it is referenced to and the dates of the SPOT 6/7 images. The reader would appreciate a simple figure indicating the timeline of mass motion events and measurements.

Precipitation analysis: Even though it takes a prominent part in the study conclusions, no details of the statistical analysis showing that the pre-event winter snow fall was outstandingly large, are given.

SPECIFIC COMMENTS

How many dGPS points were taken?

Abstract, last sentence: This statement sounds as if the 2015 Ghorka earthquake disaster can be mostly attributed to the avalanche destruction. Please rewrite.

Page 4, Line 3: Is Table 1 correct reference?

Page 4, Lines 6-7: Quantify "high elevation".

Page 4, Lines 32-33: When were the precipitation ratios observed?

Some minor typos are present.

Sometimes, the name "Yala" is followed by "Glacier", sometimes it is not. Is this a place or only a glacier? Please clarify.

Page 5, Line 1: "we corrected traditions": unclear. Quantify "huge earthquake" (magnitude and epicenter).

Section 3.1: Please make sure that the numbers exactly agree with the tables. "same magnitude" should be specified. It is sometimes not clear, which surface and initial deposit is being discussed.

Page 5, Line 26: Is "significant" appropriate in a statistical sense?

Page 5, Line 37: "7 May" –> 10 May? Also, which area is assumed in the calculation of the number given in this sentence?

Page 5, Lines 39-40: "hard to melt" is weak statement. It would help to see a number on melt reduction with respect to uncovered ice.

Page 6, Line 12: "weight" –> mass. Why is this information needed? I assume that the trajectory analysis neglected air resistance and therefore did not require the projectile mass.

Page 6, Lines 26-27: Some information on tree age measurements is needed.

Page 6, Line 28: Quantify "large".

Page 6, Line 36: "1 m" large rocks are substantial. Where does this threshold come from?

Page 6, Line 38: Rewrite "condition as of ice cliff exposure".

Page 7, Lines 19-21: Details/numbers of this calculation would be helpful.

Page 7, Lines 23-24: Please show this temperature drop.

Page 7, Line 34: I suggest deleting "obviously".

Page 8, Points 1 and 2: Here it is not clear what the estimates on ice volume refer to. Glacier ice? If so, this seems contradictory, because the authors had previously said how uncertain the estimates of glacier ice volumes are.

Page 8, Equation 1: It seems that this assumes temperate ice. For such high altitudes I expect cold ice. Does this make a substantial difference in view of available melt energy?

Page 8, Lines 23-24: This sentences needs a reference or justification.

Page 8, Line 25: How was the energy released?

Page 8, Line 29: Whose initial speed is this point referring to? The boulder's?

Page 8, Point 8: Here it seems necessary to specify uncertainties of the given numbers.

Page 8, Point 10: Where do the 2.0 m surface lowering and 1 to 2 m debris thickness come from?

Page 9, Point 10: Here it would help to state HOW the debris-covered area was reduced during the monsoon.

Page 9, 11: I suggest given numbers for the initial and lost volumes, so that the reader can confirm their origin.

Section 4.2, first paragraph: The authors discuss glacier dynamics as a potential error source. This seems unrealistic or at least unintuitive, because depending on the glacier's size, glacier dynamic reaction to seasonal mass balance changes should be negligible.

Section 4.2, third paragraph: Here the authors should better justify their assumptions, otherwise the reader has the impression that they played with numbers until they matched. Why is only the no-winter-melt prone source area taken into account? Can snow also be mobilized outside such an area? What is the reason for choosing exactly 1/2*900 kg m^-3 for snow density?

FIGURES

Include summit elevations in all photos/maps.

[Figure]

Figure 2: Parts of the legend and the GPS location are difficult to read/find.

Figure 3: What are do the thick contour lines represent?

Figure 5: There are various regions of elevation change, which are not commented on (e.g., far Eastern and far Western regions).

Figure 7: I cannot confirm the "rock existence".

Figure 8: Is this figure really needed? How does this help to determine tree age?

Figure 9: It is hard to distinguish the thin, thick and blue lines in Panel d.

REFERENCES

Podolskiy, E. A., Nishimura, K., Abe, O., & Chernous, P. A. (2010). Earthquake-induced snow avalanches: II. Experimental study. Journal of Glaciology, 56(197), 447-458.

---

## Author Comment (AC1) · 3 Mar 2017

**Dear Reviewer #1,**

Thank you for reading and commenting our manuscript, although in most cases we do not agree with the criticism brought forward. We provide our replies point by point below. Comments and replies are shown in *italic* and **bold**, respectively.

*The authors present interesting data and hypothesis on the disastrous ice avalanche event triggered by the Gorkah earthquake in the Langtang valley.*

*However the paper mixes up facts measured by different methods (weather stations, photogrammetric DSMs, statistics) with quite bold hypothesis on the process which are not underlay sufficiently by facts and measurements. The authors have to state clearly which stamens are based on facts and which statements are hypothesis or guesses. Right now these two types of stamens are dangerously mixed up.*

[reply] **We clearly distinguish observational facts from speculations. We describe the observed facts (DEMs and their differences, move of the Cheese boulder, diameter of fallen trees, and so on) in the results sections and rather speculative issues in the discussion sections. Although the estimate of ignition speed for the "Cheese boulder" is based on multiple assumptions, we have included it in the results sections. We clearly describe how we estimate ignition speed using a range of plausible scenarios (P6L15).**

*I do not agree on the hypothesis of the avalanche with only marginal contribution of glacier ice. The letter by Lacroix (2016), which is cited by the authors, clearly shows that considerable mass of up to 30 m thickness detached close to the ridge at several locations. This must be glacier ice and cannot be snow (estimated snow thickness 1.5 m). So the major part of the mass hate to come from glacier ice. The whole part on the triggering and the dynamics of the event is very weak. Based on the presented data no sound standing explanation of the triggering and dynamics of the event is possible. The authors should therefore delete or at least substantially reduce this part and declare it as hypothesis.*

[reply] **We clearly describe that the elevation difference between two DEMs estimated by Lacroix (2016) is reliable (P9L13). But we further describe in detail (P9L13-) that the use of an initial image taken in April 2014 is problematic. During one year, which includes the summer monsoon season of 2014, the glacier surface is likely to have changed significantly by melting, accumulation and glacier flow. We emphasize**

therefore that the difference of the DEMs with one year in between is NOT equivalent to "the detached ice".

Moreover, the reviewer seems to suggest that because the detached ice is thicker (30m) than the snow pack (1.5 meter), most mass must be ice. As we have shown with our volume estimates this is obviously not true, because the snow covered area (8.73 km$^2$, P9L31) is much large than the area of detached ice (< 0.5 km$^2$, which is a rough but with ArcGIS estimate by us because the area is not provided in Lacroix (2016)).

*As far as I know David Breashears also scoured high resolution photographs of the upper part of the area. Why are these photographs not used to generate a DSM?*

[reply] We appreciate this suggestion. We have generated an orthomosaic for the upper part from photographs taken by David Breashears. However, the quality of the orthomosaic is not sufficient to investigate the avalanche source area above 6000 m a.s.l. though topographic feature around glacier terminus around 5000 m a.s.l. is clear. We suppose that this insufficient quality is due to 1) upward viewing angle of the photographs out of a helicopter and 2) less contrast on snow covered areas, which is a common issue on remotely sensed DEM creation. We will add some descriptions for this problem in the revised manuscript.

*Also the story about the Chees Boulder is very vague. I agree that it is very interesting but I do not think that the facts are sound enough to include it into a scientific publication.*

[reply] As we described in the discussion paper, this is not only an interesting story but additional evidence that constrains estimates of speed and energy of the avalanche. Assumptions and limitations of this evidence are discussed in the manuscript.

*In my opinion parts of the presented data is interesting. However in the presented form the paper is not acceptable for publication in a scientific journal and has to be carefully overworked by clearly dividing measured facts and reducing speculations not supported by sufficient measurements or observations.*

[reply] The Langtang disaster represents a significant geophysical event with tremendous human consequences. Our analysis combines measured facts and interpretation, and is thus similar to other hazard studies. This particular review comment does not provide any specific details on how the manuscript should be improved, and so we have no particular response to add.

---

## Author Comment (AC2) · 3 Mar 2017

**Dear Dr. Fabian Walter,**

Thank you for your thoughtful comments, which are surely helpful to improve our study. We provide our replies point by point below. Comments and replies are shown in *italic* and **bold**, respectively.

*The submitted manuscript by Fujita et al. documents the extent of the earthquake-induced Alpine mass motion (snow/ice avalanches and rock falls), which destroyed and covered the village of Langtang (Nepal) as a result of the 2015 Gorkha earthquake. Combining differential GPS measurements after the rock fall/avalanche events with digital elevation models derived from satellite and airborne (helicopter and UAV) measurements, the authors estimate a volume of the debris cover and its evolution over time. This constitutes one of the two central messages, which this study provides. For the second message, the authors use meteorological data and further imagery to suggest source constraints, namely that the primary avalanche damage was triggered by glacier collapse and unusual pre-event winter precipitation magnified the extent of the destruction.*

*The paper is clearly written and presents a valuable assessment of a tragic natural disaster. The provided geographic constraint on debris cover seems somewhat detached from the discussion of the avalanche source. However, since the Gorkha earthquake and its damage are still relatively recent events, it should be OK to present such different aspects in a single paper.*

*My main criticism deals with several assumptions, which the authors seem to make but do not fully justify. Most of all, did the increased pre-event winter snowfall really make a significant difference? On Page 4, Line 13 the authors state that for the region of concern, only 20-30% of the annual totals in snow accumulation are attributed to winter snow fall. Does this mean that similar or much larger avalanches are typically to be expected after the monsoon season? Do the authors assume that winter snow cover is particularly vulnerable to earthquake shaking?*

**[reply] Generally speaking, summer temperature is too high for precipitation to fall as snow even at the elevation where glaciers exist. In my mass balance simulation for Yala Glacier (Fujita and Nuimura, 2011) using similar dataset used in this study, the equilibrium line altitude (ELA), at which the accumulated snow should be melted away at the end of summer, fluctuates around 5300-5400 m a.s.l., and has risen in**

recent decades. Below the ELA, no snow can be expected at the end of summer season. In addition, steep topography of the upper catchment in this study has to be taken into account. Some of the fallen snow in summer should be relocated to lower elevation by small but frequent avalanches, and the snow should be melted away at the lower elevations. During our field experiences since the 1980s, we have heard from villagers that avalanches from Mt. Langtang Lirung occasionally occurred in small tributary valleys between Langtang Village and Kyangjin (once a couple of years) though we cannot provide statistical evidence of volume and/or frequency. However, no such event was reported for the Langtang main village so far. During the winter between 2014 and 2015, snow at high elevation should be relocated with avalanche and accumulated somewhere above 5000 m a.s.l., at which a large terrace could catch the avalanches and the snow could not be melted during the winter. We briefly address this issue (summer snow does not matter) in the revised manuscript.

*Additional explanations and/or specifications of assumption are necessary for the subtracting of DEM's, the discussion of the hanging glacier collapse and calculation of avalanche source volume. Revisions should not require additional analysis. However, in its current state, the manuscript is not entirely clear on how the conclusions are reached. Result presentation and figure quality could also be improved, but these are likely more minor points.*

*MAIN REMARKS*
*Extreme snow fall: As mentioned above, it is not clear why the winter snowfall, even though it was particularly strong, amplified the avalanche devastation significantly. This seems to imply that either snow avalanche risk in in spring 2015 was larger than during the rest of the year and/or the earthquake was more likely to happen after winter than after summer (when most of the snowfall occurs). Neither of these points is argued for.*
[reply] We have replied to this comment above, and agree that the snow avalanche risk was greater in spring 2015. Summer snow cannot cause huge avalanches, and earthquake occurrence is independent of season.

*As a first step, the time series in Figures 9a-c should be extended to a full year to show if solid winter precipitation dominated solid summer precipitation.*
[reply] Our paper argues that the snowfall during the winter of 2014-2015 was anomalous, not that winter precipitation was greater than summer precipitation.

Summer snow fall and snowline elevations are strongly affected by summer temperature and less negative vertical temperature gradients. Based on observed station temperatures, we can simply assume that all winter snow accumulates (i.e. does not melt) at elevations above 5000 m a.s.l.

*Next, a discussion about earthquake-triggered avalanches would be helpful. Podolskiy et al. (2010 in Journal of Glaciology) conclude that during earthquake shaking, avalanche failure occurs primarily along weak layers or the base of snow samples. Is such triggering more likely during certain seasons? Alternatively, if the trigger was entirely due to glacier collapse, is this more likely during winter?*

[reply] Without detailed snowpack measurements it is difficult to assess the stability of the snowpack. From our station records, multiple heavy snowfalls occurred in mid-October, early January, and early March (Figure 9). Given the high solar radiation totals at the site, it is possible that sun crusts formed on the surface of these large snowfall events. The formation of a faceted layer on top of the sun crust may result in a weakness that could lead to significant slab avalanches (Jamieson, 2006), and these would be more likely during the spring when snowpacks are at their maximum. Any weakness in the snowpack would have likely been activated by the intense shaking, but a glacier collapse could be an additional trigger.

Unfortunately, Podolskiy et al. (2010) does not provide any information for the high Himalayas. We think that it is difficult to quantify the contributions of glacier collapse and winter snow with the available data and images. So in the revised manuscript, we weaken our assertion about the dominant contribution of winter snowfall though we do not withdraw our assertion that the use of SPOT6/7-DEMs is not appropriate to estimate volume of collapsed glacier. If this sentence asks literally "glacier collapse", we have no idea nor data tgagkglo evaluate it.

Jamieson, B., 2006. Formation of refrozen snowpack layers and their role in slab avalanche release. Reviews of Geophysics, 44(2), doi:10.1029/2005RG000176

*Finally, it would help to see an estimate of how much avalanche-prone snow volume is usually available for these kinds of events, excluding extreme snowfalls such as documented in the manuscript.*

[reply] This event literally has no precedent, and the anecdotal evidence we have for smaller avalanches in the region is insufficient for the estimation of typical snow avalanche volumes.

*DEM Generation: This discussion inevitably involves lots of numbers. Unfortunately, the authors sometimes round numbers, and sometimes they do not. Also, standard deviation is not always indicated. This makes it very hard and tedious for the reader to flip between Tables 2,3 and the text. I suggest more consistency.*

**[reply] We now provide consistent values in both Tables and text with standard deviation throughout the manuscript.**

*Furthermore, I could not follow the reasoning behind the "initial deposit" (Line 15 on Page 5). I thought the off-debris area was deposit-free? I guess that fresh debris deposits would have been noticed during the GPS survey. Is it reasonable to believe the large bias given the similarly large standard deviation? I am not an expert on DEM comparison, but I expect that some statistical argument is needed here.*

**[reply] We did not identify any deposit on the off-debris area during the GPS survey. We believe that the statistical evaluation for our DEMs was adequately performed with Figs. 3 and 4.**

*Projectile motion of boulder: I did not understand how the initial launch speed of the boulder relates to the avalanche. The authors seem to argue that the air blast (I suggest clearly defining this term) ahead of the avalanche launched the boulder. However, on Line 19 of Page 6 they compare boulder speed with "speed of muddy flow". In any case, there should be some explanation on moment transfer between avalanche debris or air and boulder. Is there really a simple explanation why the different speeds should be the same?*

**[reply] We use the term "air blast" not based on the avalanche dynamics (so-called snow cloud), but based on villagers' statements. We agree that more dense "muddy flow" (dense flow in the avalanche dynamics) should blow the Cheese boulder. We use the consistent term "muddy flow" with a definition in the revised manuscript.**

*Glacier trigger: The authors provide evidence for a glacier collapse that could have triggered the avalanche. They should specify if this is the only collapse that occurred in the time window of interest or not. In addition, one of the conclusions of this study seems to be that glacier ice did not exceed a few percent of the avalanche volume. In contrast, on Line 38 of Page 6 they say that their ice cliff samples may not be representative. Generally, I suggest being clearer on the glacier collapse trigger, because this is a poorly understood topic in glaciology. To my understanding, it is not generally*

*clear how glacier and underlying bedrock failure play a part in large-scale avalanches.*

**[reply] We admit that Fig. 12 in the discussion paper is not sufficient to conclude the little contribution of glacier collapse. Many other portions, which are not identified in the distorted image of the Google Earth, could have been collapsed. As we replied to the reviewer #1, we have tried to create the 3D image for the upper glacier area from the helicopter oblique photographs but could not generate good product because of 1) upward viewing angle of the photographs out of a helicopter and 2) less contrast on snow covered areas, which is a common issue on remotely sensed DEM creation. We withdraw the use of Fig. 12 and weaken our original assertion that winter snowfall is the dominant source of the avalanche and little contribution of collapsed glacier ice.**

*Presentation of geographic and timing information: The manuscript would benefit from more altitude information. All maps/photos should include altitudes of mountain peaks, villages and perhaps other sites such as glacier tongues. Currently, it is hard to put the glacier collapse in context without knowing its altitude with respect to the rest of the avalanche source area. Moreover, what are the approximate glacier volumes? In section 4.2, the authors state that the collapsed glacier cliff has a similar volume to all of Yala Glacier. This seems very small. I furthermore suggest specifying the earthquake date whenever it is referenced to and the dates of the SPOT 6/7 images. The reader would appreciate a simple figure indicating the timeline of mass motion events and measurements.*

**[reply] We add altitudes information in Fig. 1 as much as possible. We use NOT "volume" BUT "thickness" of Yala Glacier to estimate a volume of collapsed ice because no other information is available. This may be a simple misunderstanding. Anyhow, this will not be used in the revised manuscript because we withdraw the use of Fig. 12 and the estimate of collapse volume. We provide three dates of the earthquake, and of SPOT images.**

*Precipitation analysis: Even though it takes a prominent part in the study conclusions, no details of the statistical analysis showing that the pre-event winter snow fall was outstandingly large, are given.*

**[reply] We do not understand this comment. We addressed clearly what PDFs were tested with the Kolmogorov–Smirnov test, and then the Gumbel PDF was selected. Does this comment mean that we have to provide all parameter values and KS-scores for the tested PDFs? We think that the present description is sufficient.**

*SPECIFIC COMMENTS*

*How many dGPS points were taken?*

**[reply] It is practically impossible to provide number of dGPS points, which was measured at 1-sec interval, because we walked not only in the domain analyzed in this study but also along Langtang Valley. In addition, the number of measured points is often meaningless because, even if we took rest and we put the antenna on the ground, the GPS kept logging the position at 1-sec interval. Therefore, we provide the cell number of GPS-derived 1-m DEM in Fig. 4b (n = xxxx), which depends on the calculated domain. We do not change the related description.**

*Abstract, last sentence: This statement sounds as if the 2015 Ghorka earthquake disaster can be mostly attributed to the avalanche destruction. Please rewrite.*

**[reply] We did not intend to suggest this. The last sentence has been rewritten as follows: "Considering long-term observational data, probability density functions, and elevation gradients of precipitation, we conclude that this anomalous winter snow was an extreme event with a return interval of at least 100 years. The anomalous winter snowfall may have amplified the disastrous effects induced by the 2015 Gorkha earthquake in Nepal."**

*Page 4, Line 3: Is Table 1 correct reference?*

**[reply] This should be Table 2. We have corrected it.**

*Page 4, Lines 6-7: Quantify "high elevation".*

**[reply] Although it is difficult to quantify a threshold, we change here as "elevation higher than 5500 m a.s.l.".**

*Page 4, Lines 32-33: When were the precipitation ratios observed?*

**[reply] We add "in 2012/2013" and "in 1985/1986".**

*Some minor typos are present.*

*Sometimes, the name "Yala" is followed by "Glacier", sometimes it is not. Is this a place or only a glacier? Please clarify.*

**[reply] We add "Glacier" to "Yala" at only one place in P4L32. The other "Yala" without "Glacier" are found in the figure captions (Figs 1 and 9). These two denote not glacier but AWS sites "near Yala Glacier" so that we do not change them.**

*Page 5, Line 1: "we corrected traditions": unclear. Quantify "huge earthquake" (magnitude and epicenter).*

**[reply] This is mistake of "collected". We now add information for the previous earthquake.**

*Section 3.1: Please make sure that the numbers exactly agree with the tables. "same magnitude" should be specified. It is sometimes not clear, which surface and initial deposit is being discussed.*

**[reply] We correct the values in the text to keep consistency with the tables.**

*Page 5, Line 26: Is "significant" appropriate in a statistical sense?*

**[reply] We replace this by "remarkable".**

*Page 5, Line 37: "7 May" –> 10 May? Also, which area is assumed in the calculation of the number given in this sentence?*

**[reply] Yes, this should be 10 May. We correct it. We considered the deposition on the off-debris area to estimate the volume only on 7 May. We add the notation about it.**

*Page 5, Lines 39-40: "hard to melt" is weak statement. It would help to see a number on melt reduction with respect to uncovered ice.*

**[reply] It is impossible to provide any exact value without measuring thickness and thermal property of debris mantle. This is a general description so that we do not change here.**

*Page 6, Line 12: "weight" –> mass. Why is this information needed? I assume that the trajectory analysis neglected air resistance and therefore did not require the projectile mass.*

**[reply] We replace "weight" by "mass". The mass of the boulder is not required to estimate the initial speed. But we think that this information should be provided as a part of the field observation for future analysis. To avoid misunderstanding, we add descriptions that the mass is not necessary to estimate the initial speed but estimated for the future analysis.**

*Page 6, Lines 26-27: Some information on tree age measurements is needed.*

**[reply] We change here, from "35 years, which suggests that" to "35 years by**

**counting the tree ring. It suggests that".**

*Page 6, Line 28: Quantify "large".*
**[reply] We add "(> 1200 mm a⁻¹, greater than +1.4σ)". This is a threshold for the largest fourth annual accumulation during the 30-year reconstructed record from the neighboring Dasuopu ice core, which was shown in the literature cited.**

*Page 6, Line 36: "1 m" large rocks are substantial. Where does this threshold come from?*
**[reply] We do not understand what "substantial" means. Does this mean "very large"? or "important information"? It was impossible to measure directly the rocks embedded in ice cliffs or in ice tunnels because of difficult/dangerous accessibility. So this is rough estimate. We cannot provide theoretical threshold for this. We do not change here.**

*Page 6, Line 38: Rewrite "condition as of ice cliff exposure".*
**[reply] We have rewritten this line as "Thickness of the debris mantle on the ice is 0.90 ± 0.87 m (Fig. 8b) however the thickness varies widely so these estimates may not be representative values due to the limited sample number and condition of the ice cliff exposures. "**

*Page 7, Lines 19-21: Details/numbers of this calculation would be helpful.*
**[reply] We add "warmer by 0.5 °C", "–5.8 °C km⁻¹" for the lapse rate, and "(–7.6 °C for the four months from December to March)" for the winter temperature at Yala Glacier.**

*Page 7, Lines 23-24: Please show this temperature drop.*
**[reply] We do not catch what the reviewer suggests. It is shown in Fig. 9d as we cite in the main text. But anyhow, for easier understanding, we add an arrow in the figure.**

*Page 7, Line 34: I suggest deleting "obviously".*
**[reply] We delete it.**

*Page 8, Points 1 and 2: Here it is not clear what the estimates on ice volume refer to. Glacier ice? If so, this seems contradictory, because the authors had previously said how uncertain the estimates of glacier ice volumes are.*
**[reply] We do not address any source of this deposit. This is deposited volume, which**

is the sum of debris-covered area (6.55 ± 1.07 million m$^3$) and off-debris area. This is addressed in the section 3.2. We add description for the additional deposit in Table 3.

*Page 8, Equation 1: It seems that this assumes temperate ice. For such high altitudes I expect cold ice. Does this make a substantial difference in view of available melt energy?*
**[reply] We estimate this by assuming specific heat of ice (2090 J kg$^{-1}$) and its temperature at the high elevation (–7.6 °C as averaged air temperature at Yala Glacier AWS for the 2014/2015 winter). The estimated water content is reduced by 0.7% (from 7.3% to 6.7%). We do not think that this is substantial difference. We briefly address this in the revised manuscript.**

*Page 8, Lines 23-24: This sentences needs a reference or justification.*
**[reply] This sentence follows the previous description. To weaken the description, we replace "Most" by "Some amount of" and add "muddy flow".**

*Page 8, Line 25: How was the energy released?*
**[reply] The energy should be used for melting snow and ice, air blast, and muddy flow. It is impossible to quantify the energy distributions. We add the description.**

*Page 8, Line 29: Whose initial speed is this point referring to? The boulder's?*
**[reply] We add "of the boulder".**

*Page 8, Point 8: Here it seems necessary to specify uncertainties of the given numbers.*
**[reply] We add "± 0.92" for volume and "± 1.58" for surface elevation change.**

*Page 8, Point 10: Where do the 2.0 m surface lowering and 1 to 2 m debris thickness come from?*
**[reply] Lowering comes from Table 3. We add uncertainty "± 1.02" for surface elevation change and "± 0.58" for volume. We add "based on the elevated surface by the rock fall between 8 and 10 May as addressed above" for debris thickness increased with the additional event on 12 May.**

*Page 9, Point 10: Here it would help to state HOW the debris-covered area was reduced during the monsoon.*
**[reply] This is mainly due to change in surface condition on the opposite slope. We add the description.**

*Page 9, 11: I suggest given numbers for the initial and lost volumes, so that the reader can confirm their origin.*

**[reply] We provide initial and melted volumes.**

*Section 4.2, first paragraph: The authors discuss glacier dynamics as a potential error source. This seems unrealistic or at least unintuitive, because depending on the glacier's size, glacier dynamic reaction to seasonal mass balance changes should be negligible.*

**[reply] Here we do not address glacier response to the seasonal mass exchange. In general, glacier surface in the accumulation area tends to be lowered to compensate the accumulated snow thickness. It is understandable if we imagine a steady state glacier, which keep the glacier surface at a given elevation. We believe that the present description is good enough to be understood. We do not change the description.**

*Section 4.2, third paragraph: Here the authors should better justify their assumptions, otherwise the reader has the impression that they played with numbers until they matched. Why is only the no-winter-melt prone source area taken into account? Can snow also be mobilized outside such an area? What is the reason for choosing exactly 1/2\*900 kg m^-3 for snow density?*

**[reply] We have a temperature record at Yala Glacier, which guarantees "no snow melting through the 2014/2015 winter". So that we limit the source area above 5000 m a.s.l. (elevation of the Yala Glacier site is 4830 m a.s.l.). We believe that the description "Limiting the source area…no snow melt can be assumed in winter (Fig. 9c)" fully addressed it. We calculate the winter snow density from our two records of snow depth and cumulative precipitation at Yala Glacier as 385 ± 9 kg m$^{-3}$ and the recalculate the snow thickness and initial volume as 1.82 ± 0.46 m and 15.89 ± 4.05 ×10$^6$ m$^3$. We replace "surprisingly" by "reasonably".**

*FIGURES*

*Include summit elevations in all photos/maps.*

**[reply] We add contour lines and summit elevation (only for Mt. Langtang Lirung) in Fig. 1 but not for others. Because Figs 2 and 5 contain so much information, we do not want to make them more complex.**

*Figure 2: Parts of the legend and the GPS location are difficult to read/find.*

**[reply] We enlarge the legend. We do not understand what the GPS location means. This is not a specific site but measurement points. Because the GPS measurement was conducted at 1-sec interval, it looks "black lines". We add description in the figure caption.**

*Figure 3: What are do the thick contour lines represent?*

**[reply] We do not understand what this comment means. We describe that the thick contour lines denote elevation bias. We have no idea how we change the expression so that we do not change here.**

Figure 5: There are various regions of elevation change, which are not commented on (e.g., far Eastern and far Western regions).

**[reply] We add the descriptions which area we point out.**

*Figure 7: I cannot confirm the "rock existence".*

**[reply] This is what we want to show. We replace "existence" by "absence".**

*Figure 8: Is this figure really needed? How does this help to determine tree age?*

**[reply] Yes, this is necessary indeed. Although we do not use tree diameter for tree age counting (we do not describe so), the diameter of fallen trees will be supportive data for estimating the impact of the air blast and muddy flow (larger tree, more impact and vice versa). Debris thickness is also supportive information to quantify the contributions of snow and rock avalanches. Although we do not quantify them in the present study, the information will be helpful if some researchers want to analyze them. We can move this figure into the supplementary material but it should be determined by the editor.**

*Figure 9: It is hard to distinguish the thin, thick and blue lines in Panel d.*

**[reply] Even if green and blue are difficult to be distinguished, "shadings" help to distinguish the lines. In addition, the temperature drop after the earthquake, which is the main message of this panel, is now pointed by an arrow. We want to keep consistency between colors and sites in the figure so that we do not change here.**

---

## Author Response (AR1)

**Dear Editor,**

Thank you for handling our manuscript and your thoughtful comments. We provide our replies point by point below. Comments and replies are shown in *italic* and **bold**, respectively. In the revised manuscript, the added and revised parts are colored by red. Figures in the manuscript are moderate resolution. Those with high resolution will be provided when this is accepted.

*My greatest concerns are with the quality of the figures. For example, I think that Fig 1 could be improved by making it more "three-dimensional". Perhaps it would be better to make two figures out of Fig. 1: You obviously need an overview figure of the entire region -- but the blue polygon describing the runout area lacks detail. It does not show the steep slope above the deposition area. Would it be possible to add a figure with a close-up of Langtang village, including a two-dimensional profile of the avalanche track? This would give the interested reader a better overview of the terrain and the location of the deposits. And the travel distance is immense -- I think some 5km to 7km.*
**[reply] We add three 3D views such as overview including Mt. Langtang Lirung, close-up of the debris covering the village and elevation difference. All figures include extent of debris and village facility. [Fig. 1b, Fig. 12]**
**We do not add avalanche track because 1) this is out of focus of this study, 2) it should be multiple possible tracks, 3) but we have no evidence (so too speculative), 4) it is easily obtainable from freely available DEM and satellite image on GIS software so that anyone can make it if they want, and 5) Lacroix (2016) provides possible profiles in his Figure 5c.**

*For me, the most important Figures are 2 and 5. In these images it is very difficult to see where the avalanche came from. These images should also demarcate not only the location of the debris -- but also the destruction caused by the avalanche air blast. (That is, the location of the debris relative to the villages. I know the tie-points are of interest to the authors--but I, as a modeller, am primarily interested in the limits of the destruction. For me Figs. 2 and 5 are also overloaded. I think that Figure 5a should be placed in a separate figure so that the reader can imagine how large the measured deposits are! Imagine standing in front of 20 m high deposits! This is unique data and should be highlighted.*
**[reply] We add three 3D close-up views of the debris covering the village and elevation difference including extent of debris and village facility. [Fig. 12]**

*In general, I think there are two few pictures giving the reader an overview of the terrain and destruction -- Perhaps the photos in Fig. 7 should be related to a map. Where are they exactly?*

**[reply] We add locations of tree investigation and ice cliffs. [Fig. 2a]**

*Fig 8 mixes tree diameter and debris thickness. True, both are histograms, but they are completely different quantities. The debris thickness I would place nearer to Fig. 5, the elevation difference.*

**[reply] We separate the histogram figure. [Figs. 6 and 9]**

*You state that the ice "should have played a key role in initiating the entire event".*

**[reply] We add this phrase. Thanks. [L409]**

*Would it be possible to place a table in the paper with your estimated mass balance: (Mass/Volume of initial ice and snow, Mass/Volume of entrained snow and debris; mass/volume of dirt/rocky debris. I think this would be helpful for the reader to identify what kind of event it was -- ice-avalanche, snow avalanche or debris avalanche. It would also provide valuable constraints on avalanche dynamics models.*

**[reply] We reevaluate the volumes and add Table 4. Thanks. [L379-, Table 4]**

*Furthermore, it would highlight the role of entrainment -- or did the entire slope become instable at once from the shaking?*

**[reply] It is totally unknown. We add our excuse. [L382]**

*Figure 9: It is hard to distinguish the thin, thick and blue lines in Panel d.*

**[reply] We change line style. [Fig. 10d]**